



# Diagnosing the sensitivity of grounding line flux to changes in sub-ice shelf melting

Tong Zhang[1], Stephen F. Price[1], Matthew J. Hoffman[1], Mauro Perego[2], and Xylar Asay-Davis[1]

[1]Fluid Dynamics and Solid Mechanics Group, Los Alamos National Laboratory, Los Alamos, NM, 87545, USA
[2]Center for Computing Research, Sandia National Laboratories, Albuquerque, NM, 87185, USA
**Correspondence:** T. Zhang (tzhang@lanl.gov), S. Price (sprice@lanl.gov)

**Abstract.** We seek to understand causal connections between changes in sub-ice shelf melting, ice shelf buttressing, and grounding-line flux. Using a numerical ice flow model, we study changes in ice shelf buttressing and grounding line flux due to localized ice thickness perturbations – a proxy for changes in sub-ice shelf melting – applied to idealized (MISMIP+) and realistic (Larsen C) domains. From our experiments, we identify a correlation between a locally derived buttressing number on

the ice shelf, based on the first principal stress, and changes in the integrated grounding line flux. The origin of this correlation, however, remains elusive from a physical perspective; while local thickness perturbations on the ice shelf (thinning) generally correspond to local increases in buttressing, their integrated impact on changes at the grounding line are exactly the opposite (buttressing at the grounding line decreases and ice flux at the grounding line increases). This and additional complications encountered when examining realistic domains motivates us to seek an alternative approach, an adjoint-based method for

calculating the sensitivity of the integrated grounding line flux to local changes in ice shelf geometry. We show that the adjoint-based sensitivity is identical to that deduced from pointwise, diagnostic model perturbation experiments. Based on its much wider applicability and the significant computational savings, we propose that the adjoint-based method is ideally suited for assessing grounding line flux sensitivity to changes in sub-ice shelf melting.

## 1   Introduction

Marine ice sheets like that overlying West Antarctica (and to a lesser extent, portions of East Antarctica) are grounded below sea level and their bedrock would remain so even after full isostatic rebound (Bamber et al., 2009). This and the fact that ice sheets generally thicken inland leads to a geometric configuration prone to instability; a small increase in flux at the grounding line thins the ice there leading to floatation, a retreat of the grounding line into deeper water, further increases in flux (due to still thicker ice), and further thinning and grounding line retreat. This theoretical "marine ice sheet instability"

(MISI) mechanism (Mercer, 1978; Schoof, 2007) is supported by idealized (e.g., Schoof, 2012; Asay-Davis et al., 2016) and realistic (e.g., Cornford et al., 2015; Royston and Gudmundsson, 2016) ice sheet modeling experiments, and some studies (Joughin et al., 2014; Rignot et al., 2014) argue that such an instability is currently under way for outlet glaciers of Antarctica's Amundsen Sea Embayment (ASE). The relevant perturbation for grounding line retreat in the ASE is thought to be intrusions of relatively warm, intermediate-depth ocean waters onto the continental shelves, which have reduced the thickness and extent





of marginal ice shelves via increased sub-ice shelf melting (e.g., Jenkins et al., 2016). These reductions are important because fringing ice shelves restrain the flux of ice across their grounding lines farther upstream – the so-called "buttressing" effect of ice shelves (Gudmundsson et al., 2012; Gudmundsson, 2013; De Rydt et al., 2015) – which makes them a critical control on the rate of ice flux from Antarctica to the ocean.

On ice shelves, the driving stress (from ice thickness gradients) is balanced by gradients in longitudinal stress (Hutter,
1983; Morland, 1987; Schoof, 2007) and an ice shelf in one horizontal dimension ($x,z$) provides no buttressing (Schoof, 2007; Gudmundsson, 2013). For realistic, three-dimensional ice shelves, however, buttressing results from three main sources: 1) along-flow compression, 2) lateral shear, and 3) "hoop" stress (Wearing, 2016). Compressive and lateral shear stresses can provide resistance to extensional ice shelf flow through along- and across-flow stress gradients. The less commonly discussed "hoop" stress is a transverse stress arising from azimuthal extension in regions of diverging flow (Pegler and Worster, 2012;
Wearing, 2016). Due to the complex geometries, kinematics, and dynamics of real ice shelves, an understanding of the specific processes and locations that control ice shelf buttressing is far from straightforward.

Several recent studies apply whole-Antarctic ice sheet models, optimized to present-day observations, towards improving our understanding of how Antarctic ice shelves impact ice dynamics farther upstream or limit flux across the grounding line. Fürst et al. (2016) proposed a locally derived "buttressing number" (extended from Gudmundsson, 2013) for Antarctic ice
shelves and used it to guide the location of calving experiments whereby the removal of progressively larger portions of the shelves near the calving front identified dynamically "passive" ice shelf regions; removal of these regions (e.g., via calving) was found to have little impact on ice shelf dynamics or the flux of ice from ice upstream to the calving front. Reese et al. (2018) conducted a set of forward model perturbation experiments to link small, localized decreases in ice shelf thickness to changes in integrated grounding-line flux (GLF), thereby providing a map of GLF sensitivity to local increases in sub-ice shelf
melting.

Motivated by these studies, we build on and extend the methods and analysis of Fürst et al. (2016) and Reese et al. (2018) to address the following questions: (1) Do local evaluations of ice shelf buttressing reflect how local perturbations in ice shelf thickness impact grounding line flux[1]? (2) What are the limitations of locally derived buttressing metrics when used to assess GLF sensitivity? (3) Can new methods overcome these limitations? Our specific goal is to identify robust methods for
diagnosing where on an ice shelf changes in thickness (here, assumed to occur via increased sub-ice shelf melting) have a significant impact on flux across the grounding line. Our broader goal is to contribute to the understanding of how increased sub-ice shelf melting can be expected to impact the dynamics and stability of real ice sheets.

Below, we first provide a description of the ice sheet model used in our study and the model experiments performed. We then analyze and discuss the experimental results in order to quantify how well easily evaluated, local buttressing metrics correlate
with modeled changes in GLF. This leads us to propose and explore an alternative, adjoint-based method for assessing GLF sensitivity to ice shelf thickness perturbations. We conclude with a summary discussion and recommendations.

---

[1]For example, are the local evaluations of buttressing from Fürst et al. (2016) related to the GLF changes modeled by Reese et al. (2018)?



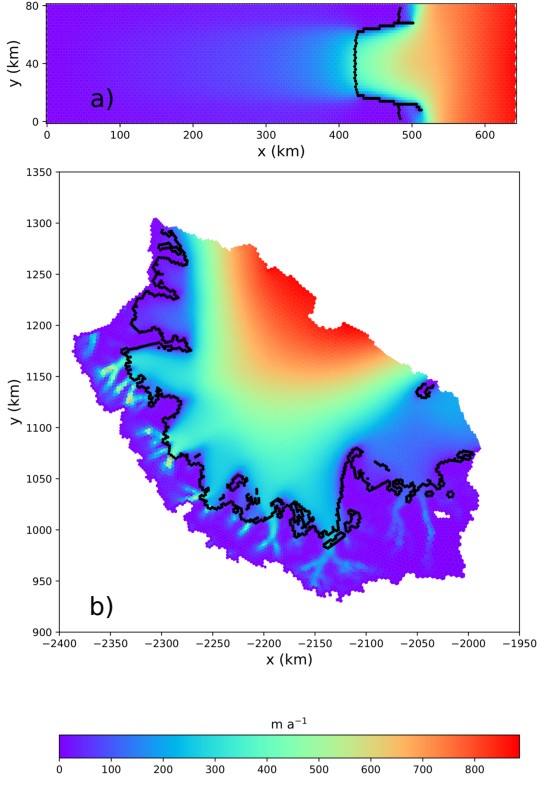

**Figure 1.** (a) Plan view of surface speed for the MISMIP+ and (b) Larsen C Ice Shelf experimental domains. For the Larsen C domain, velocities have been optimized to match observations from Rignot et al. (2011). Black curves indicate the location of the grounding line.

## 2   Model description

We use the MPAS-Albany Land Ice model (MALI; Hoffman et al., 2018), which solves the three-dimensional, first-order approximation to the Stokes momentum balance for ice flow. Using the notation of Perego et al. (2012) and Tezaur et al. (2015a) this can be expressed as,

$$
\begin{cases}
-\nabla \cdot (2\mu_e \dot{\boldsymbol{\epsilon}}_1) + \rho_i g \frac{\partial s}{\partial x} &= 0, \\
-\nabla \cdot (2\mu_e \dot{\boldsymbol{\epsilon}}_2) + \rho_i g \frac{\partial s}{\partial y} &= 0,
\end{cases}
\tag{1}
$$

where $x$ and $y$ are the horizontal coordinate vectors in a Cartesian reference frame, $s(x,y)$ is the ice surface elevation, $\rho_i$ represents the ice density, $g$ the acceleration due to gravity, and $\dot{\boldsymbol{\epsilon}}_{1,2}$ are given by

$$
\dot{\boldsymbol{\epsilon}}_1 = \left( \begin{array}{ccc} 2\dot{\epsilon}_{xx} + \dot{\epsilon}_{yy}, & \dot{\epsilon}_{xy}, & \dot{\epsilon}_{xz} \end{array} \right)^T,
\tag{2}
$$

and

$$
\dot{\boldsymbol{\epsilon}}_2 = \left( \begin{array}{ccc} \dot{\epsilon}_{xy}, & \dot{\epsilon}_{xx} + 2\dot{\epsilon}_{yy}, & \dot{\epsilon}_{yz} \end{array} \right)^T.
\tag{3}
$$



The "effective" ice viscosity, $\mu_e$ in Eq. (1), is given by

$$\mu_e = \gamma A^{-\frac{1}{n}} \dot{\epsilon}_e^{\frac{1-n}{n}}, \tag{4}$$

where $\gamma$ is an ice stiffness factor, $A$ is a temperature-dependent rate factor, $n = 3$ is the power-law exponent, and the effective
strain rate, $\dot{\epsilon}_e$, is defined as

$$\dot{\epsilon}_e \equiv \left( \dot{\epsilon}_{xx}^2 + \dot{\epsilon}_{yy}^2 + \dot{\epsilon}_{xx}\dot{\epsilon}_{yy} + \dot{\epsilon}_{xy}^2 + \dot{\epsilon}_{xz}^2 + \dot{\epsilon}_{yz}^2 \right)^{\frac{1}{2}}, \tag{5}$$

where $\dot{\epsilon}_{ij}$ are the corresponding strain-rate components.

Under the first-order approximation to the Stokes equations, a stress free upper surface can be enforced through

$$\dot{\boldsymbol{\epsilon}}_1 \cdot \mathbf{n} = \dot{\boldsymbol{\epsilon}}_2 \cdot \mathbf{n} = 0, \tag{6}$$

where $\mathbf{n}$ is the outward pointing normal vector at the ice sheet upper surface, $z = s(x,y)$. The lower surface is allowed to slide
according to the continuity of basal tractions,

$$2\mu_e \dot{\boldsymbol{\epsilon}}_1 \cdot \mathbf{n} + \beta u = 0, \quad 2\mu \dot{\boldsymbol{\epsilon}}_2 \cdot \mathbf{n} + \beta v = 0, \tag{7}$$

where $\beta$ is a spatially variable friction coefficient, $2\mu_e \dot{\boldsymbol{\epsilon}}_{1,2}$ represent the viscous stresses, and $\boldsymbol{u}$ is the two-dimensional velocity
vector $(u, v)$. On lateral boundaries in contact with the ocean, the portion of the boundary above sea level is stress free while
the portion below sea level feels the ocean hydrostatic pressure according to

$$2\mu_e \dot{\boldsymbol{\epsilon}}_1 \cdot \mathbf{n} = \tfrac{1}{2}\rho_i g H \left( 1 - \tfrac{\rho_i}{\rho_w} \right) n_1, \quad 2\mu_e \dot{\boldsymbol{\epsilon}}_2 \cdot \mathbf{n} = \tfrac{1}{2}\rho_i g H \left( 1 - \tfrac{\rho_i}{\rho_w} \right) n_2, \tag{8}$$

where $\mathbf{n}$ is the outward pointing normal vector to the lateral boundary (i.e., parallel to the $(x,y)$ plane), $\rho_w$ is the density of
ocean water, and $n_1$ and $n_2$ are the $x$ and $y$ component of $\mathbf{n}$. A more complete description of the MALI model, including the
implementations for mass and energy conservation, can be found in Hoffman et al. (2018). Additional details on the momentum
balance solver can be found in Tezaur et al. (2015a, b).

Here, we apply MALI to experiments on both idealized and realistic marine-ice sheet geometries. For our idealized domain
and model state, we start from the equilibrium initial conditions for the MISMIP+ experiments, as described in Asay-Davis
et al. (2016). For our realistic domain, we use Antarctica's Larsen C Ice Shelf and its upstream catchment area. For the
Larsen C domain, the model state is based on the optimization of the ice stiffness ($\gamma$ in Eq. (4)) and basal friction ($\beta$ in Eq.
(7)) coefficients in order to provide a best match between modeled and observed present-day velocities (Rignot et al., 2011)
using adjoint-based methods discussed in Perego et al. (2014) and Hoffman et al. (2018). The domain geometry is based on
Bedmap2 (Fretwell et al., 2013), and ice temperatures, which are used to determine the flow factor and held fixed for this
study, are taken from Liefferinge and Pattyn (2013). Mesh resolution on the ice shelf is between 2 and 4 km and coarsens to
5 km in the ice sheet interior. Following optimization to present-day velocities, the model is relaxed using a 100-year forward
run, providing the initial condition from which the Larsen C experiments are conducted (as discussed below). The domain
and initial conditions were extracted from the Antarctica-wide configuration used by MALI for initMIP experiments (Hoffman



et al., 2018; Seroussi et al., 2019). Both the MISMIP+ and Larsen C experiments use 10 vertical layers that are finest near the bed and coarsen towards the surface. The grounding line position is determined from hydrostatic equilibrium. A sub-element parameterization is used to define basal friction coefficient values at the grounding line (Seroussi et al., 2014).

## 3 Perturbation experiments

To explore the sensitivity of changes in GLF to small, localized changes in ice shelf thickness, we conduct a number of perturbation experiments analogous to those of Reese et al. (2018). Using diagnostic model solutions, we calculate the instantaneous response of GLF for the idealized geometry and initial state provided by the MISMIP+ experiment (Asay-Davis et al., 2016). We then conduct a similar study for Antarctica's Larsen C Ice Shelf using a realistic configuration and initial state. The geometry and steady-state ice speeds for MISMIP+ and Larsen C Ice Shelf are shown in Fig. 1.

Our experiments are conducted in a manner similar to those of Reese et al. (2018). We perturb the coupled ice sheet-shelf system by decreasing the ice thickness uniformly by 1 m over grid cells (or square boxes containing a number of grid cells) covering the base of the ice shelves, after which we examine the instantaneous impact on kinematics and dynamics (discussed further below). For MISMIP+, we use a uniform hexagonal mesh with a horizontal resolution of around 2 km and we perturb the thickness at single cells in the mesh. For the Larsen C Ice Shelf, horizontal mesh resolution is spatially variable and – to maintain consistency with the experiments of Reese et al. (2018) – we assign each grid cell to fall within one and only one $20\times20$ km square perturbation "box", to which thickness perturbations are applied uniformly. Lastly, for the MISMIP+ 2-km experiments, we note that, in order to save on computing costs, we only perturb the region of the ice shelf for which $x < 530$ km (the area over which the ice shelf is laterally buttressed) and for which $y > 40$ km (due to symmetry about the center line). We do, however, analyze the response to these perturbations over the entire model domain.

Similar to Reese et al. (2018), we define a GLF response number for our perturbation-based experiments,

$$N_{rp} = \frac{R}{P}, \tag{9}$$

where $R$ is the change in the ice (mass) flux integrated along the entire grounding line due to a perturbation in the thickness at a single grid cell (or box of grid cells in the case of Larsen C), and $P$ is the local mass change associated with the perturbation. The subscript $rp$ denotes the "response" from "perturbation" experiments[2]. Note that $N_{rp}$ is dimensionless.

Distal changes in GLF (quantified by $N_{rp}$) in response to a local change in ice shelf thickness are assumed to occur via changes in ice shelf buttressing, which generally acts to resist the flow of ice across the grounding line. To quantify the local ice shelf buttressing capacity, we calculate a dimensionless buttressing number, $N_b$, analogous to that from Gudmundsson (2013) and Fürst et al. (2016),

$$N_b\left(\mathbf{n}\right) = 1 - \frac{\mathrm{T_{nn}}}{N_0}, \tag{10}$$

---

[2]To distinguish from other approaches to be discussed below.





where $\mathrm{T}_{nn} := \mathbf{n} \cdot \mathbf{Tn}$ is a scalar measure of the stress normal to the surface defined by $\mathbf{n}$. $N_0$ is the value that $\mathrm{T}_{nn}$ would take if

the ice was removed up to the considered location and replaced with ocean water[3], and it is defined as $N_0 := \frac{1}{2}\rho_i\left(1 - \rho_i/\rho_w\right)gH$,

with $\rho_i$ and $\rho_w$ being the densities of ice and ocean water, respectively[4]. While Gudmundsson (2013) chose the unit vector $\mathbf{n}$

to be normal to the grounding line to define the "normal" buttressing number, Fürst et al. (2016) extended his definition to the

ice shelf by examining $N_b\left(\mathbf{n}\right)$ for $\mathbf{n}$ along the ice flow direction and along the direction of the second principal stress. Here, we

explore the connection between changes in grounding line flux (quantified by $N_{rp}$), sub-shelf melting, and local buttressing on

the ice shelf (quantified by $N_b$[5]) corresponding to arbitrary $\mathbf{n}$ (in order to consider all possible relationships on the ice shelf).

Below, we refer to $\mathrm{T}_{nn}$ as the "normal stress". The tensor $\mathbf{T}$ is defined as follows, based on the two dimensional shallow shelf

approximation[6],

$$
\mathbf{T} = \left\{ \begin{array}{cc} 4\mu_e\dot{\epsilon}_{xx} + 2\mu_e\dot{\epsilon}_{yy} & 2\mu_e\dot{\epsilon}_{xy} \\ 2\mu_e\dot{\epsilon}_{yx} & 4\mu_e\dot{\epsilon}_{yy} + 2\mu_e\dot{\epsilon}_{xx} \end{array} \right\}. \tag{11}
$$

We elaborate further on the calculation of the buttressing number in Appendix A. Additional details of ice shelf dynamics

based on the shallow shelf approximation can be found in Greve and Blatter (2009).

## 4   Results

### 4.1   Correlation between buttressing and changes in GLF

A decrease in ice shelf buttressing tends to lead to an increase in GLF (e.g., Gagliardini et al., 2010, also see Fig. 2a) and

intuitively we expect that the GLF should be relatively more sensitive to ice shelf thinning in regions of relatively larger

buttressing. We aim to better understand and quantify the relationship between the local ice shelf buttressing "strength" in

a given direction (characterized by $N_b$) and changes in GLF (characterized by $N_{rp}$). A reasonable hypothesis is that, for a

given ice thickness perturbation, the resulting change in the GLF is proportional to the buttressing number at the perturbation

location.

In Fig. 2, we show the results from all (730) perturbation experiments for MISMIP+, and the corresponding $N_{rp}$ and $N_b$

values. For $N_b$, we show the values corresponding to three different directions, corresponding to the choice of $\mathbf{n}$ in Eq. (10);

the first principal stress direction ($\mathbf{n}_{p1}$), the second principal stress direction ($\mathbf{n}_{p2}$), and the ice flow direction ($\mathbf{n}_f$). In the

discussion below, we frequently refer to these three directions when discussing the buttressing number. In agreement with the

findings of Fürst et al. (2016), the largest values for $N_b$ occur when it is calculated in the $\mathbf{n}_{p2}$ direction. While there appears

to be a qualitatively reasonable spatial correlation between the magnitude of $N_{rp}$ and $N_b$ when the latter is calculated in the

---

[3]Or alternatively, the resistance provided by a static, neighboring column of floating ice at hydrostatic equilibrium.

[4]For the MISMIP+ experiment, $\rho_i$=918 kg m$^{-3}$ and for the Larsen C experiment, $\rho_i$=910 kg m$^{-3}$. For both experiments, $\rho_w$=1028 kg m$^{-3}$.

[5]Note that we do not discuss the tangential buttressing number defined by Gudmundsson (2013)). Hereafter, we use "buttressing number" to refer exclusively to the "normal buttressing number", as defined above.

[6]While we employ a three-dimensional, first-order Stokes approximation here (Tezaur et al., 2015a), the depth-varying and depth-averaged solutions converge to the same value on ice shelves, where basal resistance is zero.

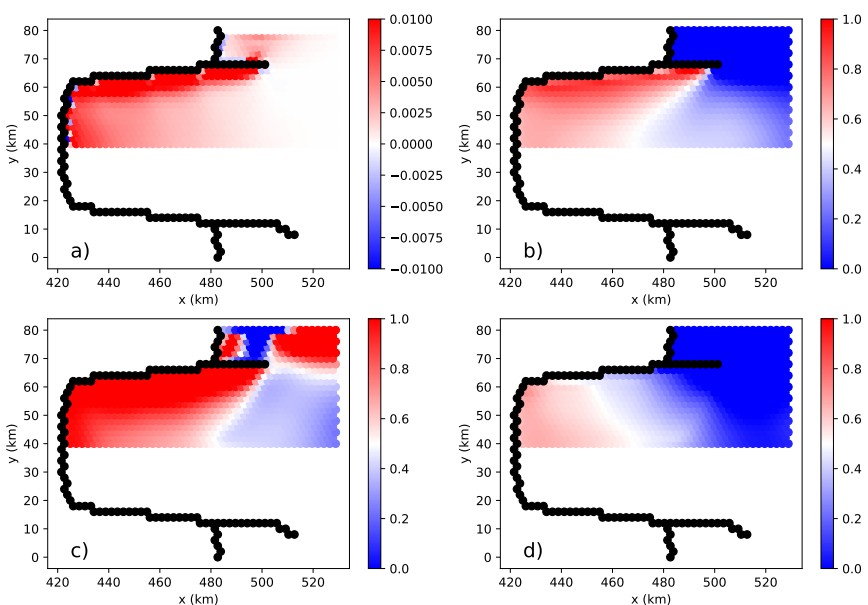

**Figure 2.** The 730 perturbation points for the MISMIP+ experiments. (a) The spatial distribution of the GLF response number, $N_{rp}$. (b-d) The spatial distribution of the buttressing number, $N_b$, corresponding to directions (b) $\mathbf{n}_{p1}$, (c) $\mathbf{n}_{p2}$, and (d) $\mathbf{n}_f$. Black dots indicate grid cells located along the grounding line. The negative $N_{rp}$ values in (a) correspond to a few partially-grounded cells in the vicinity of the GL, where the GLF can be reduced by ice shelf thinning. Here the colorbars for (a)–(d) do not show the full data range.

$\mathbf{n}_{p1}$ and $\mathbf{n}_{p2}$ directions (and less so when calculated in the $\mathbf{n}_f$ direction), in Fig. 3 we show that there is no clear relationship between the response number $N_{rp}$ and the buttressing number $N_b$ calculated along any of these directions, at least for the case where we consider all points on the ice shelf. In Fig. 4, however, we show correlations (Figs. 4b–d) between the modeled value of $N_{rp}$ and $N_b$ where we have removed points for which the flow is weakly buttressed ($x > 480\,\text{km}$, where the ice shelf starts to become unconfined) and where the minimum distance to the grounding line is less then 12 km (Fig. 4a). In this case, stronger, near-linear $N_{rp} : N_b$ relationships emerge. In particular, a stronger correlation between $N_{rp}$ and $N_b$ occurs when $N_b$ is calculated using $\mathbf{n}_{p1}$ (Fig. 4b), relative to when using $\mathbf{n}_{p2}$ (Fig. 4c) or $\mathbf{n}_f$ (Fig. 4d).

### 4.2 Directional dependence of buttressing

The buttressing number at any perturbation point depends on $\text{T}_{nn}$, which in turn depends on the chosen direction of the normal vector, $\mathbf{n}$ (Eq. 10). Fürst et al. (2016) calculated $N_b$ using $\mathbf{n}_f$ and $\mathbf{n}_{p2}$ and chose the latter – the direction corresponding to the second principal stress (the maximum compressive stress or the least extensional stress) – to quantify the local value of "maximum buttressing" on an ice shelf. In Fig. 5a, we plot the linear-regression correlation coefficients ($r$) for the $N_{rp} : N_b$

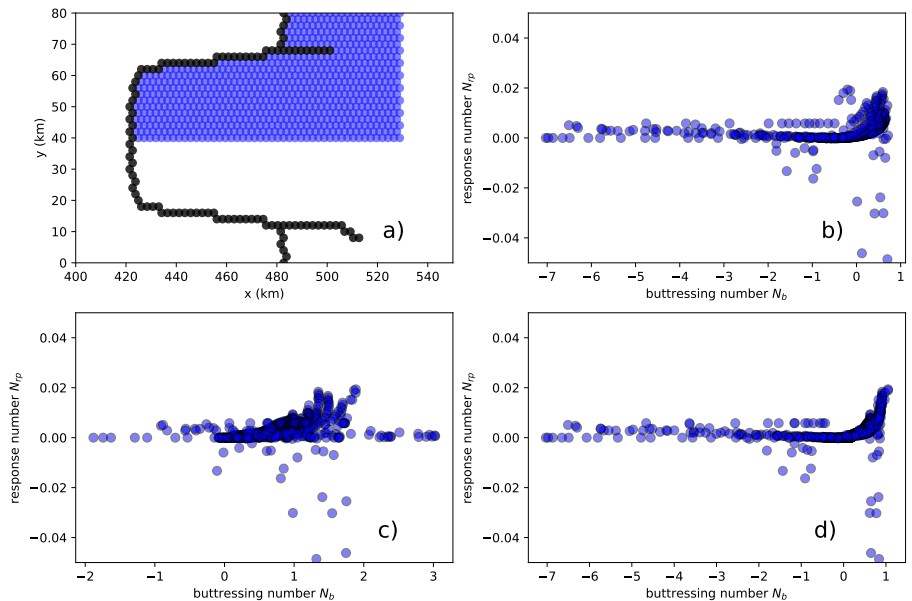

**Figure 3.** (a) Blue dots represent the locations of all perturbation points analyzed (730) for the $N_{rp} : N_b$ linear regression analysis. Black dots indicate grid cells located along the grounding line. (b-d) Modeled $N_{rp}$ from perturbation experiments versus predicted $N_{rp}$ as a function of $N_b$ calculated along (b) $\mathbf{n}_{p1}$, (c) $\mathbf{n}_{p2}$, and (d) $\mathbf{n}_f$.

relationship where the direction of $\mathbf{n}$ used in the calculation of $\mathrm{T}_{nn}$ varies continuously from $\Delta\phi =$ 0–180° relative to $\mathbf{n}_{p1}$ (we

also show how the buttressing number $N_b$ varies according to direction, starting from $\mathbf{n}_{p1}$, in Fig. S1). We find large correlation coefficients ($r > 0.9$) when $N_b$ is aligned closely with $\mathbf{n}_{p1}$ ($\Delta\phi = 0°$ or $180°$) and the smallest correlation coefficient ($r < 0.5$) when $N_b$ is aligned with $\mathbf{n}_{p2}$ ($\Delta\phi = 90°$). Similar conclusions can be reached when examining the continuous values for $r$ with respect to the ice flow direction (Fig. 5b), where correlations are phase shifted by approximately 50° counter-clockwise relative to Fig. 5a. Clearly, the best correlation occurs along the direction between $\mathbf{n}_{p1}$ and $\mathbf{n}_f$. Note that we do not see an exact match

between Fig. 5a and Fig. 5b if we shift the angle by 50° because the angular difference between $\mathbf{n}_{p1}$ and $\mathbf{n}_f$ varies slightly between individual grid cells where thickness perturbations are applied (distributions for the angular difference between $n_{p1}$ and $n_f$ are shown in Fig. S2).

    Fürst et al. (2016) posit that $N_b(\mathbf{n}_{p2})$ provides the best buttressing metric and chose it for identifying regions of maximum buttressing on an ice shelf. Here, however, we find that, compared to $N_b(\mathbf{n}_{p2})$, $N_b(\mathbf{n}_{p1})$ and $N_b(\mathbf{n}_f)$ show a better correlation

with changes in GLF via local, sub-ice shelf melt perturbations. We return to and discuss these differences further below in Section 5.


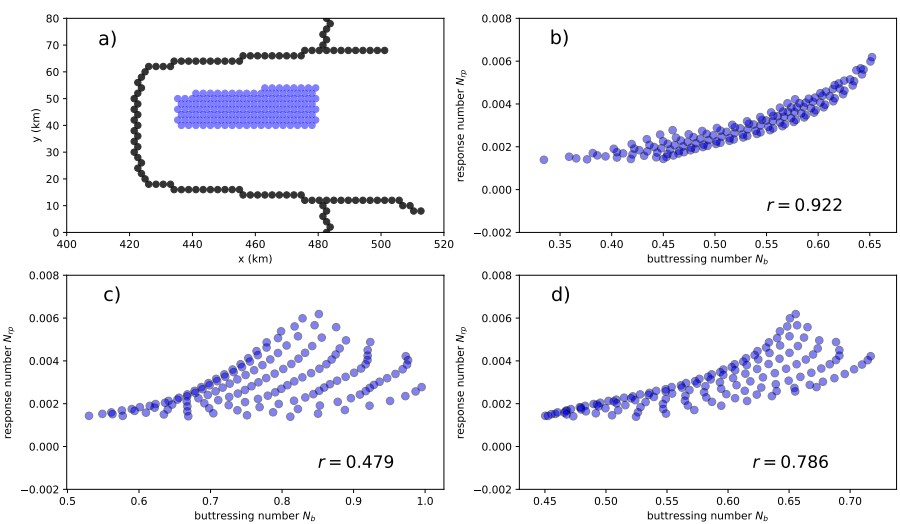

**Figure 4.** (a) Blue dots represent the locations of all perturbation points analyzed (142) for the $N_{rp} : N_b$ linear regression analysis. Black dots indicate grid cells located along the grounding line. (b-d) Modeled $N_{rp}$ from perturbation experiments versus predicted $N_{rp}$ as a function of $N_b$ calculated along (b) $\mathbf{n}_{p1}$, (c) $\mathbf{n}_{p2}$, and (d) $\mathbf{n}_f$. The correlation coefficient for each modeled $N_{rp}$ versus $N_b$ is given by $r$.

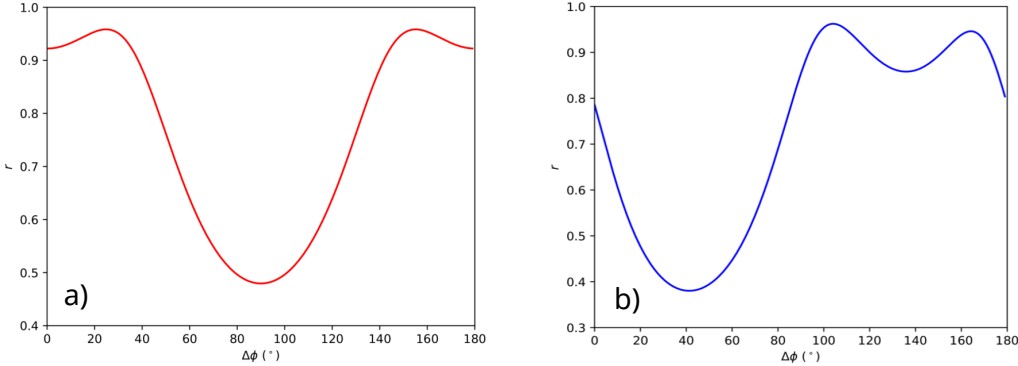

**Figure 5.** Correlation coefficients for the linear regression relationship of $N_{rp} : N_b$ where $\mathbf{n}$ is rotated counterclockwise by $\Delta\phi$ degrees relative to (a) $\mathbf{n}_{p1}$ and (b) $\mathbf{n}_f$. The perturbation points analyzed here are the same as in Fig. 4a.





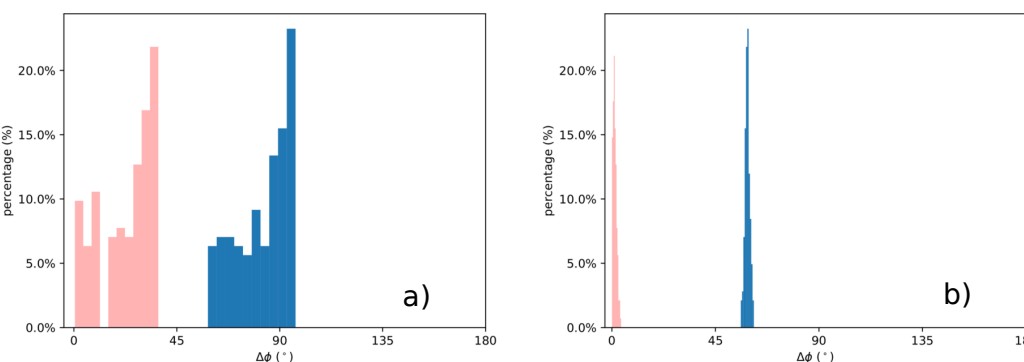

**Figure 6.** Histograms for the maximum (red) and minimum (blue) percent speed increases in grid cells adjacent to a thickness perturbation point, plotted as a function of angular distance with respect to (a) $\mathbf{n}_{p1}$ and (b) $\mathbf{n}_f$. Points analyzed are those from Fig. 4a.

## 4.3 Local, far-field, and integrated impacts of changes in buttressing

We now examine how local thickness perturbations on the ice shelf lead to local changes in geometry and velocity, and in turn, local changes in buttressing. Our aim is to better understand how perturbations affect buttressing locally (on the ice shelf) and, in turn, impact the overall buttressing and ice flux at the grounding line.

### 4.3.1 Changes in geometry, velocity, and buttressing in the vicinity of ice shelf thickness perturbations

To better understand the local impacts of a perturbation on the local ice velocity at each perturbation location, we calculate both the maximum and the minimum increase in ice speed[7] among neighboring cells (i.e., two values for each perturbation point) and the orientation of these neighboring cells relative to the $\mathbf{n}_{p1}$ and $\mathbf{n}_f$ direction at the perturbation point. The results are plotted as histograms in Fig. 6. Note that for our hexagonal mesh, there are six neighboring cells adjacent to each perturbed cell so that only a discrete number of directions (6) can be examined. The maximum and minimum speed increases cluster at 0–45° and near 60–90° relative to $\mathbf{n}_{p1}$, respectively (Fig. 6a). A similar relationship is seen in Fig. 6b, where the maximum and minimum speed increases cluster near 0° and 60°, respectively, relative to $\mathbf{n}_f$. Hence, the maximum ice speed increases near a perturbation are generally more closely aligned with the $\mathbf{n}_{p1}$ and $\mathbf{n}_f$ directions and minimum ice speed increases are more closely aligned with the $\mathbf{n}_{p2}$ direction[8]. This suggests that local ice shelf thickness perturbations induce local speed changes along a favored direction, which here aligns closely with the ice flow direction, or $\mathbf{n}_f$. This finding is supported by Gudmundsson (2003), who used an idealized ice flow model experiment to demonstrate that, following perturbations to the basal roughness or slipperiness, the group velocity of that perturbation propagates primarily along the main ice flow direction.

---

[7]Here, we use speed changes as a proxy for changes in the local ice flux near perturbation points on the ice shelf because, (1) thickness changes are minimal and (2) changes in speed can be approximately interpreted as changes in velocity because directional changes are small.

[8]Note that neighboring cells to a perturbation are distributed along discrete angles, so that there are generally not neighboring cells exactly along the $\mathbf{n}_{p2}$ direction.




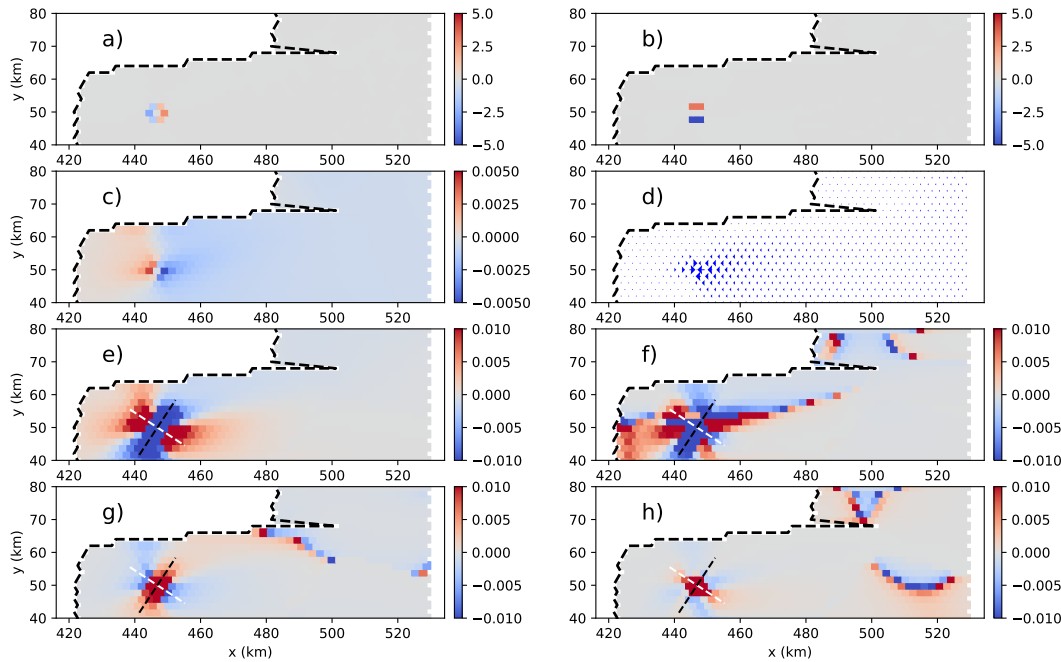

**Figure 7.** An example of the local change (ratio, in %) in (a) the ice thickness gradient in $x$, (b) ice thickness gradient in $y$, (c) ice speed, (d) ice velocity (relative), (e, f) principal strain rates, and (g, h) buttressing number following a local perturbation to the ice shelf thickness. In (e) and (g), changes (colors) are associated with the $\mathbf{n}_{p1}$ direction and for (f) and (h) changes are associated with the $\mathbf{n}_{p2}$ direction. The white- and black-dashed lines show the direction of $\mathbf{n}_{p1}$ and $\mathbf{n}_{p2}$ at the perturbation location, respectively.

Local thickness perturbations on the ice shelf alter the local ice thickness gradient; on the upstream (grounding line) side
of the perturbation, the thickness gradient will decrease and on the downstream (calving front) side it will increase (Fig. 7a,
b). Locally, the result is an increase in compression (or a decrease in extension) along both principal stress directions (Fig. 7e,
f) and, via Eq. (10), a corresponding increase in the local buttressing number calculated along both principal stress directions
(Fig. 7g, h).

Importantly, while we find that the overall spatial pattern of the change in buttressing is quite complex, variable, and depends
on the location of the perturbation, the general pattern of the local change (at or within a few grid cells of the applied pertur-
bation) is that of an *increase* in the buttressing number (see also Figs. S3 and S4 in the Supplementary, analogous to Fig. 7,
which show similar patterns but for different perturbation locations). This is further confirmed in Fig. 8, where we show that,
in response to local ice shelf thickness perturbations, for all points analyzed in Fig. 4 there is an *increase* in the buttressing
number associated with both the $\mathbf{n}_{p1}$ and $\mathbf{n}_{p2}$ directions, and for every direction in between (the larger increase in $N_b$ along
the $\mathbf{n}_{p2}$ direction emphasizes that the local ice flow is always more compressive (or less extensional) along the $\mathbf{n}_{p2}$ direction).
As we discuss next, this finding of locally increased buttressing for all perturbations applied to the ice shelf is at odds with our
desire to interpret the local buttressing number in terms of changes in GLF.




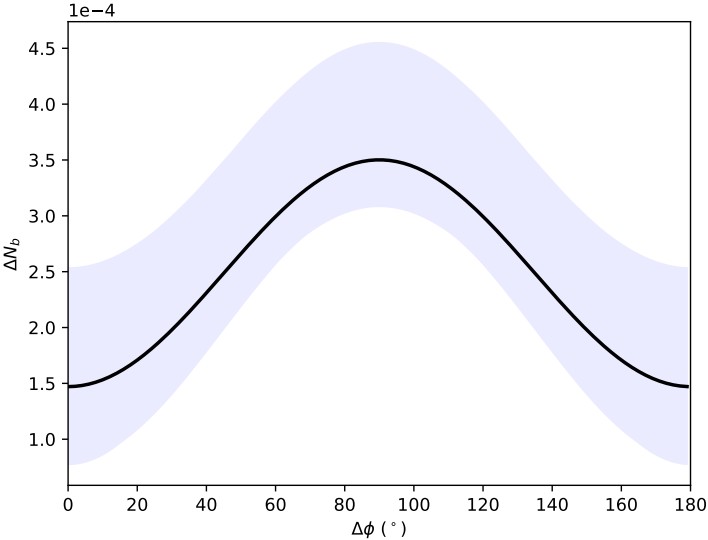

**Figure 8.** The change in buttressing number $\Delta N_b$ at the neighboring cells with maximum ice speed increase at all perturbation points. Changes in buttressing are calculated along the direction $\Delta\phi$, rotated counterclockwise relative to the $\mathbf{n}_{p1}$ direction. The points analyzed include those in Fig. 4a, which are shown as the shaded area, with the solid curve representing their mean value.

### 4.3.2  Changes in buttressing and ice flux at the grounding line

To understand how perturbations propagate across the ice shelf and impact the grounding line, we examine changes in the
buttressing number and the ice speed normal to the grounding line following local perturbations in thickness on the ice shelf. To quantify this relationship, we define $\Upsilon_{\mathrm{gl}}$,

$$\Upsilon_{\mathrm{gl}} = \mathrm{Corr}\left(\Delta\mathbf{N_b}, \Delta\mathbf{u}\right) = \frac{\mathrm{cov}\left(\Delta\mathbf{N_b}, \Delta\mathbf{u}\right)}{\sigma(\Delta\mathbf{N_b})\sigma(\Delta\mathbf{u})}, \tag{12}$$

where $\Delta\mathbf{N_b} = \mathbf{N_{bp}} - \mathbf{N_{bc}}$ and $\Delta\mathbf{u} = \mathbf{u_p} - \mathbf{u_c}$ and with the subscripts $p$ and $c$ denoting the perturbation and the "control" (i.e., the initial condition) experiments, respectively. $\Delta\mathbf{N_b}$ and $\Delta\mathbf{u}$ denote vectors of the changes in the buttressing number
and the ice speed, respectively, for all cells along the grounding line. $\Upsilon_{\mathrm{gl}}$, a correlation coefficient, is an integrated[9] measure of the consistency between the magnitude and the sign of the change in the buttressing number and ice speed between the control and perturbation experiments, with cov and $\sigma$ representing the covariance and the standard deviation, respectively.

By plotting values of $\Upsilon_{\mathrm{gl}}$ mapped to their respective perturbation locations on the ice shelf, we show that there is generally a *negative* correlation between speed and buttressing at the GL: *increases* in speed (and hence flux) across the GL correlate
with *decreases* in buttressing at the GL (Fig. 9). This negative correlation is substantially stronger when the buttressing number is calculated along the $\mathbf{n}_{p1}$ direction (Fig. 9a) than along the $\mathbf{n}_{p2}$ direction (Fig. 9b). Carrying this analysis one step further,

---

[9]Integrated in the sense that the correlation coefficient takes into account the entire grounding line.

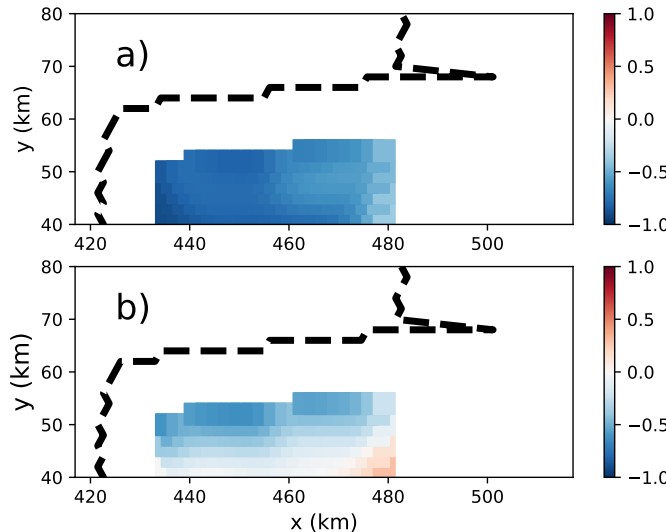

**Figure 9.** Spatial distribution of the correlation coefficient $\Upsilon_{\mathrm{gl}}$ from Eq. (12) over the MISMIP+ domain for buttressing number changes calculated parallel to (a) $\mathbf{n}_{p1}$ and (b) $\mathbf{n}_{p2}$ (colors). $\Upsilon_{\mathrm{gl}}$ is a measure of the correlation between changes in buttressing number and ice speed along the grounding line. The black-dashed line represents the grounding line, along which values of $\Upsilon_{\mathrm{gl}}$ are calculated for each perturbation on the ice shelf, as shown in Fig. 4a.

in Fig. 10 we plot $\Upsilon_{\mathrm{gl}}$ for each perturbation point (span along the $y$ axis) and for *all* directions (span along the $x$ axis) in the range of $\Delta\phi = 0\text{–}180°$ relative to $\mathbf{n}_{p1}$. Again, this correlation is generally negative and substantially stronger for buttressing numbers calculated close to the $\mathbf{n}_{p1}$ direction (i.e., for $\Delta\phi$ closer to $0°$ or $180°$).

### 4.3.3 Local versus integrated impacts of changes in buttressing

The changes in ice speed and buttressing at the grounding line that are quantified by Figs. 9 and 10 must be the result of perturbations initiated on the ice shelf that have propagated (instantaneously) to the grounding line, where increases in speed are associated with increased extension along $\mathbf{n}_{p1}$ and, according to Eq. (10), decreased buttressing associated with the $\mathbf{n}_{p1}$ direction. Intuitively, these increases in ice speed at the grounding line must be triggered by the loss of buttressing on the shelf. However, as discussed in the previous section, local perturbations and changes in buttressing on the shelf – as quantified by the changes in buttressing number calculated in all directions – are clearly not representative of the integrated changes in buttressing that are "felt" upstream at the grounding line (i.e., local increases in buttressing on the shelf versus decreases in buttressing at the grounding line). This casts doubt on the utility of assessing the GLF sensitivity using locally derived buttressing numbers on the ice shelf (a discussion we return to below). The apparent correlation between $N_{rp}$ and $N_b(\mathbf{n}_{p1})$ (Fig. 4b) may be partially explained by Figs. 9a and 10, which suggests that $\mathbf{n}_{p1}$ may be the dominant direction controlling


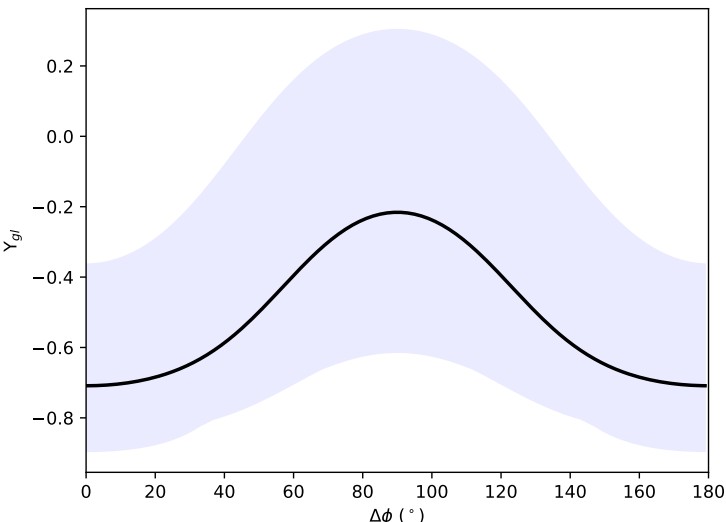

**Figure 10.** Correlation between the change in buttressing number and the change in ice speed across the grounding line (i.e., $\Upsilon_{\mathrm{gl}}$ from Eq. (12)) for the entire MISMIP+ grounding line. The horizontal axis shows how $\Upsilon_{\mathrm{gl}}$ varies as a function of the direction $\mathbf{n}$ used to define the normal stress, rotated counterclockwise from $\mathbf{n}_{p1}$ by $\Delta\phi$. Values from the maps in Figs. 9a and b plot at $\Delta\phi$ values of 0 and 90 degrees, respectively. Thus, the blue shaded region represents all possible maps for all possible values of buttressing direction. The thick black curve represents the mean value of $\Upsilon_{\mathrm{gl}}$ for any given map.

the ice flux across the grounding line for the MISMIP+ domain. In the next section, we apply a similar set of analyses to a realistic ice shelf and in doing so, demonstrate that these same correlations, already tenuous, are much more difficult to extract and interpret for realistic domains.

### 4.4 Application to Larsen C Ice Shelf

To explore whether the correlations between modeled and predicted $N_{rp}$ found for the MISMIP+ test case hold for realistic ice shelves, we apply a similar analysis to the Larsen C domain. In this case, the computational mesh resolution varies, from finer near the grounding line (2 km) to coarser towards the center of the ice shelf and calving front (4 km). In order to be comparable to the experiments and results of Reese et al. (2018), we use (approximately) 20 km $\times$ 20 km boxes for the application of ice-thickness perturbations, where the number of grid cells contained within each perturbation "box" is adjusted to sum to

their correct total area[10]. Additionally, to investigate the impacts of the complex geometry of the Larsen C Ice Shelf (i.e., the grounding line shape, the existence of ice rises, etc.), we perform two sets of perturbation experiments for which the 20 km $\times$ 20 km averaging boxes either do or do not include perturbations applied to cells near the grounding line.

---

[10]We note that the actually area may vary slightly from 400 km$^2$ depending on the number and area of the variable resolution grid cells that are included in each perturbation "box".





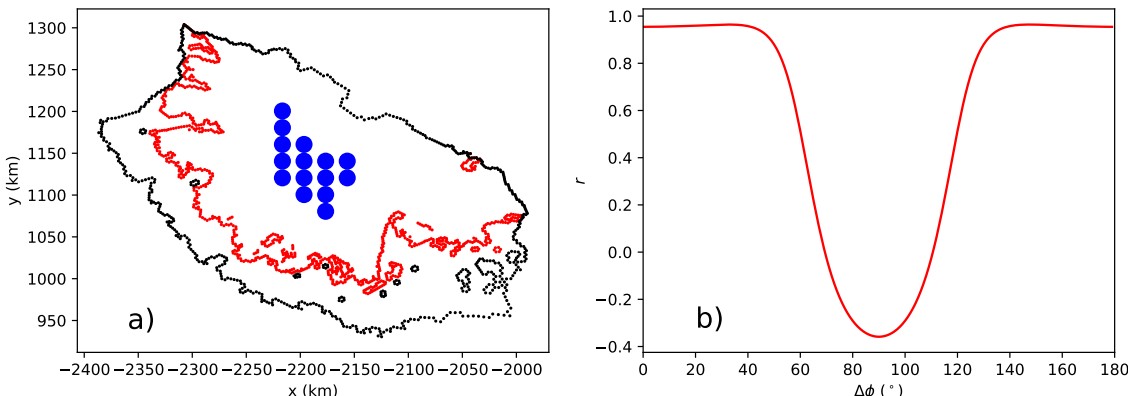

**Figure 11.** (a) The locations of the 20 km × 20 km perturbation boxes (15, solid blue dots). The red and black dotted lines are the grounding line and the boundary of model domain, respectively. (b) The $N_{rp} : N_b$ correlation coefficients for each direction rotated counterclockwise from the direction of $\mathbf{n}_{p1}$ (as in Figure 5a but for the Larsen C domain).

Analogous to Fig. 5a for the MISMIP+ test case, Fig. 11b shows the $N_{rp}{:}N_b$ correlations for the Larsen C model domain (including only perturbation points that are >50 km away from both the calving front and grounding line). As found previ-
ously, calculating $N_b$ using $\mathrm{T}_{nn}$ along $\mathbf{n}_{p1}$ provides a good overall correlation between $N_b$ and $N_{rp}$ ($\Delta\phi = 0°$ or $180°$) while calculating $N_b$ using $\mathrm{T}_{nn}$ along $\mathbf{n}_{p2}$ provides the worst overall correlation ($\Delta\phi = 90°$).

In Fig. 12 we redo the same analysis but for points nearer to the grounding line (including points that are more than 20 km away from the calving front and the grounding line). This changes the values of $r$ and also the relationship between the correlation coefficient and the alignment relative to $\mathbf{n}_{p1}$; the direction aligned with $\mathbf{n}_{p2}$ still gives the worst $N_{rp}{:}N_b$ correlation
but the direction aligned with $\mathbf{n}_{p1}$ no longer gives the best correlation. This indicates that thickness perturbations at these locations are propagating in a more complex way on a real ice shelf, especially for perturbation points that are close to the grounding line. We expect that the correlations would further degrade as additional perturbation points closer to grounding line are included in the analysis.

In the supplement, we include figures showing the the maximum and minimum ice speed increases in the vicinity of pertur-
bation cells, local changes in geometry and buttressing, and the correlations between the changes of buttressing number and ice speed on the ice shelf and along the grounding line (i.e., figures analogous to Figs. 6, 7, 8 and 10 but for the Larsen C domain instead). As for the MISMIP+ domain, the directions of maximum ice speed increase for Larsen C are also aligned closely with the ice flow direction (Fig. S5). Similarly, we find that the ice speed increase following thickness perturbations on the Larsen C shelf increase buttressing locally (Fig. S6, S7). We do not, however, find any clear correlation between changes
in buttressing number and ice speed along the grounding line for Larsen C (Fig. S8).

Overall, for a realistic ice shelf like Larsen C with a complex flow field, it is difficult to find the robust, directionally dependent relationships seen for the more idealized, MISMIP+ domain. This is likely because, for more complex and realistic





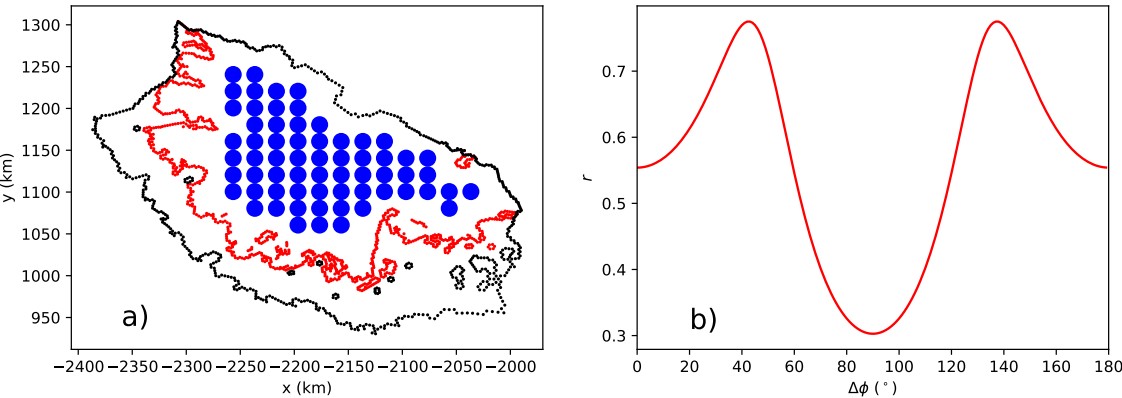

**Figure 12.** (a) The locations of the 20 km × 20 km perturbation boxes (solid blue dots). The red and black dotted lines are the grounding line and the boundary of model domain, respectively. (b) The $N_{rp} : N_b$ correlation coefficients for each direction rotated counterclockwise from the direction of $\mathbf{n}_{p1}$ (as in Figure 11b but including additional analysis points closer to the grounding line and calving front).

domains, there is no dominant direction of buttressing controlling ice flux across the grounding line. This finding further diminishes our confidence in attempting to use a simple metric like a locally derived ice shelf buttressing number. For this

reason, we now explore an alternative and more robust method for quantifying how thickness perturbations affect flux at the grounding line.

## 4.5    Adjoint sensitivity

Our goal throughout this study has been to find a simple and robust metric for diagnosing GLF sensitivity to ice shelf thickness perturbations. However, the complications discussed above suggest that this may not be possible, motivating our investigation

of a wholly different approach. This approach provides a GLF sensitivity map analogous to that provided by the Reese et al. (2018) finite perturbation-based experiments (and those conducted here). But rather than computing the GLF change due to a perturbation applied individually at each of $n$ model grid cells (thus requiring $n$ diagnostic solves), we use an adjoint-based method that allows for the computation of the sensitivity at all $n$ grid cells simultaneously for the cost of a single adjoint-model solution. Briefly, this method involves the solution of an auxiliary linear system (the adjoint system) to compute the

so-called Lagrange multiplier, a variable with the same dimensions as the forward-model solution for the ice velocity. Here, the matrix associated with the system is the transpose of the Jacobian of the first-order approximation to the Stokes flow model (Perego et al., 2012). In addition, the adjoint method requires computation of the partial derivatives of the first-order model residual and the GLF with respect to the velocity solution and the ice thickness. Here, we compute the Jacobian and all the necessary derivatives using automatic differentiation (Tezaur et al., 2015a). Additional details of the adjoint-based method and

calculations are giving in Appendix B.



A similar approach has been proposed by Goldberg et al. (2019). That work primarily assessed the adjoint sensitivity of the volume above floatation with respect to sub-ice shelf melting of Dotson and Crosson ice shelves in West Antarctica. In contrast to our approach, Goldberg et al. (2019) compute *transient* sensitivities because their quantity of interest (volume above floatation) is time dependent.

The adjoint-based sensitivity has units of mass flux per year per meter of ice thickness perturbation (kg a$^{-1}$ m$^{-1}$). We normalize this value by the mass change per year due to the thickness perturbation so that it is dimensionless and comparable to $N_{rp}$, and refer to it as $N_{ra}$ (where the subscript $a$ is for "adjoint"). In Figs. 13 and 14, we demonstrate the application of this method to the MISMIP+ and Larsen C domains by comparing GLF sensitivities deduced from 730 and 1000 individual diagnostic model evaluations (i.e., the respective perturbation experiments discussed above[11]) with those deduced from a single

adjoint-based solution. Note that for some cells adjacent to the grounding line, the negative sensitivity values may be caused by partially grounded cells (i.e., a thinning of ice thickness there may induce a decrease in ice flux across the grounding line). The comparison in Figs. 13 and 14 demonstrates that the two approaches provide a near exact match. As might be expected based on the discussion above, the two methods disagree in regions very near to the grounding line (see Fig. 15c). This discrepancy is likely a consequence of the high non-linearities near the grounding line, as suggested by the fact that the agreement between the

two methods improves as the size of the perturbation decreases (from 10 m to 0.001 m; see Fig. 15). This might be exacerbated by the sliding law adopted in this work, which results in abrupt changes in the basal traction across the grounding line (other sliding laws allowing for a smoother transition at the grounding line, e.g. Brondex et al. (2017), might mitigate this problem).

The adjoint sensitivity map represents a linearization of the GLF response to thickness perturbations. As long as the perturbations are small enough, one can approximate the GLF response by multiplying the sensitivity map by the thickness perturbation.

Comparison of $N_{ra}$ and $N_{rp}$ for different perturbation sizes (Fig. 15) suggests that this is reasonable for perturbations on the order of <10 m for points on the ice shelf that are not too close to the GL. At the same time care should be taken when interpreting the sensitivities – based on either the perturbation- or adjoint-based methods – in the vicinity of grounding lines. This is especially important when considering that the near-grounding-line region is also that with the largest sensitivities (Figs. 13a and 14a). Because these sensitivities may be inaccurate, they provide an additional argument for applying high spatial

resolution near the grounding line; coarse resolution near the grounding line will extend the region over which inaccurate sensitivites may be assessed. More accurately assessing the sensitivities near the grounding line may require the application of perturbations more realistic in both magnitude and spatial scale, as opposed to the infinitesimal, highly localized perturbations explored here.

The adjoint-method provides sensitivity maps over the entire ice shelf, including around islands, promontories, and along the

grounding line itself, which is generally the part of the ice shelf where the GLF is the most sensitive to thickness perturbations (e.g., see Figs. 13a and 14a above and Fig. 1 in Reese et al. (2018)). Thus, despite the added complexity in its computation, the adjoint-based method provides significant advantages over the simpler but more *ad hoc* analysis methods discussed above.

---

[11]For Larsen C, we conduct perturbation experiments at individual grid cells to allow for a closer comparison with the adjoint method.

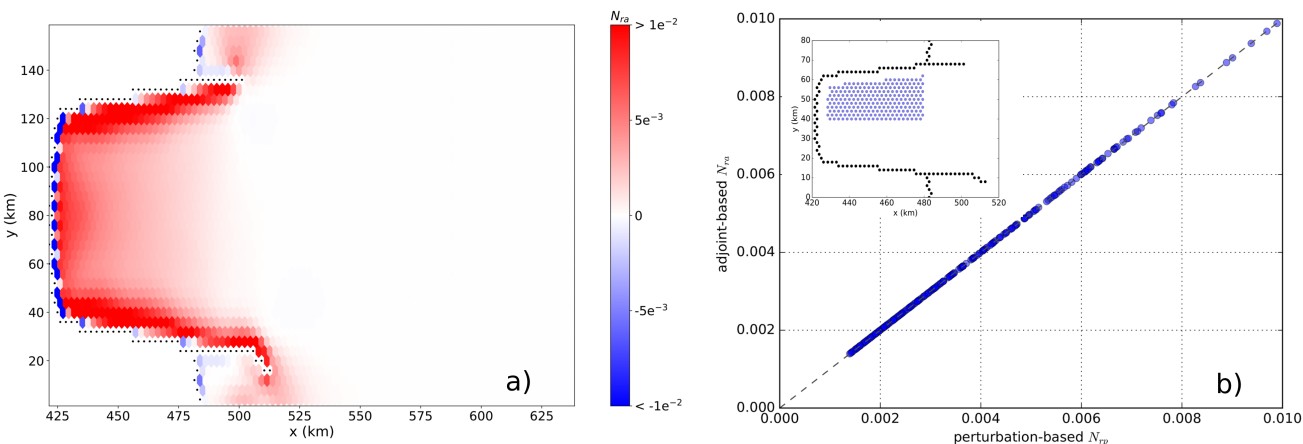

**Figure 13.** (a) Grounding line flux sensitivity for the MISMIP+ domain derived from the an adjoint model approach; (b) Perturbation- ($N_{rp}$; $x$-axis) versus adjoint-based ($N_{ra}$; $y$-axis) sensitivies are plotted against one another (perturbation locations are shown by circles in the inset, where the grounding line grid cells are shown by the black dots.)

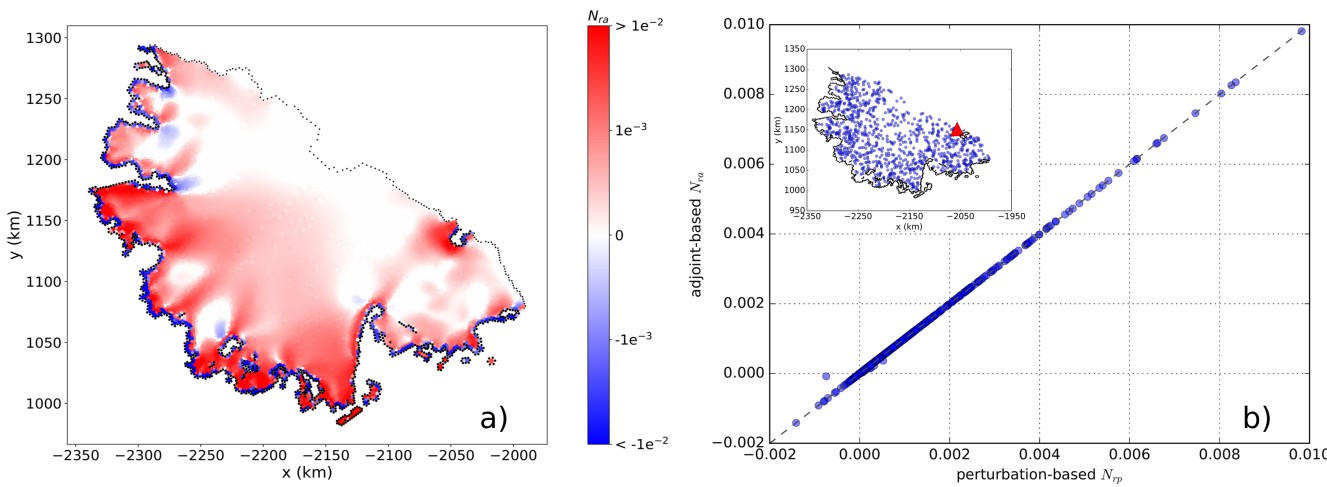

**Figure 14.** (a) Grounding line flux sensitivity for the Larsen C domain derived from the adjoint model approach; (b) Perturbation- ($N_{rp}$; $x$-axis) versus adjoint-based ($N_{ra}$; $y$-axis) sensitivies are plotted against one another (perturbation locations are shown by circles in the inset, where the one outlier in b) is at the calving front (red triangle), and the grounding line in is shown by the black curve.)



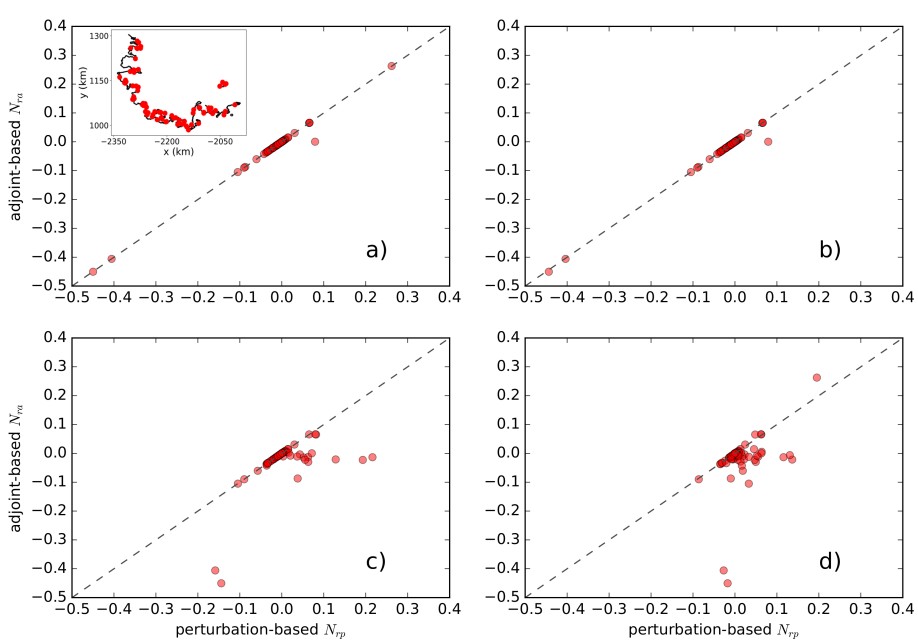

**Figure 15.** Comparisons between perturbation- and adjoint-based sensitivities ($N_{rp}$ and $N_{ra}$, respectively) for ice thickness perturbation of (a) 0.001 m, (b) 0.01 m, (c) 1 m and (d) 10 m for perturbation points near the grounding line (<3 km), as shown as the red solid circles in the inset in (a).



## 5   Discussion and Conclusions

The current interest in better understanding the controls on the MISI is due to the potential for future (and possibly present-day,
ongoing) unstable retreat of the West Antarctic ice sheet (e.g., Joughin et al., 2014; Hulbe, 2017; Konrad et al., 2018). Because
a loss of ice shelf buttressing is a primary cause of increased GLF (and thus an indirect control on the MISI), recent attention
has focused on better understanding the sensitivity of ice shelf buttressing to increases in iceberg calving and sub-ice shelf
melting. In this study, we have attempted to better characterize and quantify how local thickness perturbations on ice shelves –
a proxy for local thinning due to increased sub-ice shelf melting – impact ice shelf buttressing and GLF.

Two previously used approaches for assessing GLF sensitivity to changes in ice shelf buttressing – the flux response number
($N_{rp}$) and the buttressing number ($N_b$) – are reasonably well correlated in some situations. This correlation is, however, highly
dependent on the direction chosen to define buttressing. Specifically, we find that the choice of the normal vector used when
calculating $N_b$ dictates whether there is a general correlation or lack of correlation between $N_{rp}$ and $N_b$. Here, for both
idealized and realistic ice shelf domains, we find a stronger correlation between $N_{rp}$ and $N_b$ when the normal stress used in
calculating the buttressing number corresponds to the first principal stress or the ice flow direction, whereas the correlation is
much weaker when $N_b$ is calculated in the direction associated with the second principal stress.

These findings appear at odds with the interpretation from previous efforts of Fürst et al. (2016), who argue that buttressing
provided by an ice shelf is best quantified by $N_b$ calculated in the direction of $\mathbf{n}_{p2}$. These seemingly contradictory findings may
be partially rectified by considering the different foci of Fürst et al. (2016) versus the present work: while Fürst et al. (2016)
primarily focused on how the removal of passive shelf ice (identified by $N_b(\mathbf{n}_{p2})$) impacted ice shelf dynamics, as quantified
by the *change in ice flux across the calving front*, our focus is specifically on how localized ice shelf thickness perturbations
impact the *change in ice flux across the grounding line*[12]. While changes in calving flux are likely to impact the amount of
buttressing provided by an ice shelf, they do not directly contribute to changes in sea level. For this reason, changes in GLF are
arguably the more important metric to consider when assessing the impacts of changes in ice shelf buttressing.

Of concern in applying the apparent correlation between $N_{rp}$ and $N_b$ (relatively difficult and easy quantities to calculate,
respectively) to diagnose $N_{rp}$ from observations or models is the lack of a clear physical connection between local changes
in buttressing on the ice shelf and integrated changes in flux at the grounding line. Here, we show that localized thinning on
the shelf generally leads to a local *increase* in $N_b$. Yet these same perturbations consistently result in a *decrease* in buttressing
and an *increase* in ice flux across the grounding line. This finding suggests that local evaluations of buttressing on the ice shelf
need to be interpreted with caution, as they may not be meaningful with respect to understanding and quantifying changes in
GLF. It is also possible that the correlations we find between $N_{rp}$ and $N_b$ are simply fortuitous, and thus not meaningful in any
physical way [13].

---

[12]While Fürst et al. (2016) also discuss the impact of perturbations on the flux across the grounding line, this is a secondary focus of their paper and mostly
discussed in the Supplementary Information.

[13]We note that the correlations we discuss here, between $N_{rp}$, $N_b(\mathbf{n}_f)$, and $N_b(\mathbf{n}_{p1})$ were alluded to in the Supplement from Fürst et al. (2016), where
they note that "Outside the [Passive Shelf Ice] area, melting will affect the buttressing potential. Consequences for upstream tributary glaciers are then best
estimated from buttressing in the flow direction."


Practically speaking, however, these distinctions and concerns may be irrelevant; when realistic, complex ice shelf geometries are considered, it is not possible to define or even identify clear relationships between $N_{rp}$ and $N_b$. For the Larsen C

domain considered here, strong, positive correlations are only found to exist over a small, isolated region near the center of the ice shelf. Proximity to the grounding line, the calving front, complex coastlines, islands, and promontories all serve to degrade these correlations significantly, reducing the utility of the buttressing number as a simple metric for diagnosing GLF sensitivity on real ice shelves. Further, it is precisely these more complex regions close to the ice shelf grounding lines where sub-ice shelf thinning will result in the largest impact on changes in GLF (as demonstrated here in Figs. 13a and 14a ).

Considering these complexities, we propose that assessing GLF sensitivities for real ice shelves requires an approach much more analogous to the perturbation method used by Reese et al. (2018). Due to the computational costs and the experimental design complexity associated with the perturbation-based method we propose that an adjoint-based method is the more efficient way for assessing GLF sensitivity to changes in buttressing resulting from changes in sub-ice shelf melting. Future work should focus on applying these methods to assessing the sensitivities of real ice shelves, based on observed or modeled patterns of sub-

ice shelf melting, and assessing how these sensitivities change in time along with the evolution of the coupled ocean-and-ice shelf system.

## 6 Acknowledgements

Support for this work was provided through the Scientific Discovery through Advanced Computing (SciDAC) program funded by the U.S. Department of Energy (DOE) Office of Science, Biological and Environmental Research and Advanced Scientific

Computing Research programs. This research used resources of the National Energy Research Scientific Computing Center, a DOE Office of Science user facility supported by the Office of Science of the U.S. Department of Energy under Contract DE-AC02-05CH11231, and resources provided by the Los Alamos National Laboratory Institutional Computing Program, which is supported by the U.S. Department of Energy National Nuclear Security Administration under Contract DE-AC52-06NA25396.

## 7 Code availability

The code of the ice sheet model (MALI) can be found here: https://github.com/MPAS-Dev/MPAS-Model/releases

## 8 Author contribution

Tong Zhang and Stephen Price initiated the study with input from Matthew Hoffman, Mauro Perego, and Xylar Asay-Davis. The adjoint experiments were conducted by Mauro Perego. Tong Zhang conducted the diagnostic simulations with MALI. Tong Zhang and Stephen Price wrote the paper with contributions from all co-authors.





## 9 Competing interests

The authors declare that they have no conflict of interest.

## Appendix A: Calculation of the buttressing number

At the calving front, the stress balance is given by,

$$\sigma \cdot \mathbf{n} = -p_w \mathbf{n}, \tag{A1}$$

where $\sigma$ is the Cauchy stress tensor, $\mathbf{n}$ is the unit normal vector pointing horizontally away from the calving front, and $p_w$ is the sea water pressure. In a Cartesian reference frame, this gives two equations for the stress balance in the two horizontal directions,

$$\sigma_{xx} n_x + \sigma_{xy} n_y = -p_w n_x,$$
$$\sigma_{xy} n_x + \sigma_{yy} n_y = -p_w n_y. \tag{A2}$$

Expressing the full stress as the sum of the deviatoric stress and the isotropic pressure ($\sigma = \tau - p$) and assuming that the vertical normal stress $\sigma_{zz}$ is hydrostatic gives,

$$p = \rho_i g(s - z) - \tau_{xx} - \tau_{yy}. \tag{A3}$$

Combining Equations A2 and A3 gives,

$$(2\tau_{xx} + \tau_{yy}) n_x + \tau_{xy} n_y = -p_w n_x + \rho_i g(s - z) n_x,$$
$$\tau_{xy} n_x + (2\tau_{yy} + \tau_{xx}) n_y = -p_w n_x + \rho_i g(s - z) n_x. \tag{A4}$$

By vertically integrating Equation A4 and approximating the depth-integrated viscosity as $\overline{\mu} = \mu H$, we obtain

$$(2\tau_{xx} + \tau_{yy}) n_x + \tau_{xy} n_y = \frac{1}{2} \rho_i g (1 - \frac{\rho}{\rho_w}) H n_x,$$
$$\tau_{xy} n_x + (2\tau_{yy} + \tau_{xx}) n_y = \frac{1}{2} \rho_i g (1 - \frac{\rho}{\rho_w}) H n_y. \tag{A5}$$

If we define the two-dimensional stress tensor $\mathbf{T}$ as,

$$\mathbf{T} = \left\{ \begin{array}{cc} 2\tau_{xx} + \tau_{yy} & \tau_{xy} \\ \tau_{xy} & 2\tau_{yy} + \tau_{xx} \end{array} \right\}, \tag{A6}$$

then we will have the buttressing number ($N_b$) as

$$N_b = 1 - \frac{\mathbf{n} \cdot \mathbf{T} \mathbf{n}}{N_0}, \tag{A7}$$

where

$$N_0 = \frac{1}{2} \rho_i g (1 - \frac{\rho_i}{\rho_w}) H. \tag{A8}$$




## Appendix B: Adjoint calculation of GLF sensitivity

The adjoint method is often used to compute the derivative (or "sensitivity") of some quantity (here, the GLF) that depends on the solution of a partial differential equation, with respect to parameters (here, the ice thickness) (see, e.g., Gunzburger (2012)). It is particularly effective when the number of parameters is large because it only requires the solution of an additional linear system, independent of the number of parameters. In the discrete case, the GLF is a function of the ice speed vector, $\mathbf{u}$, and the ice thickness vector, $\mathbf{H}$. Using the chain rule, we compute the total derivative of the GLF with respect to the ice thickness as:

$$\frac{d(\text{GLF})}{d\mathbf{H}} = \frac{\partial(\text{GLF})}{\partial\mathbf{u}} \frac{\partial\mathbf{u}}{\partial\mathbf{H}} + \frac{\partial(\text{GLF})}{\partial\mathbf{H}}. \tag{B1}$$

Here $\frac{\partial\mathbf{u}}{\partial\mathbf{H}}$ denotes the matrix with components $\left(\frac{\partial\mathbf{u}}{\partial\mathbf{H}}\right)_{ij} = \frac{\partial\mathbf{u}_i}{\partial\mathbf{H}_j}$. Similarly $\frac{\partial(\text{GLF})}{\partial\mathbf{u}}$ and $\frac{\partial(\text{GLF})}{\partial\mathbf{H}}$ are row vectors with components $\frac{\partial(\text{GLF})}{\partial\mathbf{u}_j}$ and $\frac{\partial(\text{GLF})}{\partial\mathbf{H}_j}$ repsectively. In order to compute $\frac{\partial\mathbf{u}}{\partial\mathbf{H}}$, we write the finite element discretization (Tezaur et al. (2015b)) of the flow model (Eq. (1)) in the residual form $\mathbf{c}(\mathbf{u},\mathbf{H}) = \mathbf{0}$ and differentiate with respect to $\mathbf{H}$:

$$\mathbf{0} = \frac{d\mathbf{c}}{d\mathbf{H}} = \frac{\partial\mathbf{c}}{\partial\mathbf{u}} \frac{\partial\mathbf{u}}{\partial\mathbf{H}} + \frac{\partial\mathbf{c}}{\partial\mathbf{H}}. \tag{B2}$$

Here $J := \frac{\partial\mathbf{c}}{\partial\mathbf{u}}$ is a square matrix referred to as the Jacobian. It follows that $\frac{\partial\mathbf{u}}{\partial\mathbf{H}}$ is solution of

$$J\frac{\partial\mathbf{u}}{\partial\mathbf{H}} = -\frac{\partial\mathbf{c}}{\partial\mathbf{H}}. \tag{B3}$$

Note that this corresponds to solving many linear systems, one for each column of $\frac{\partial\mathbf{u}}{\partial\mathbf{H}}$ (i.e. for each entry of the ice thickness vector). We can then compute the sensitivity as

$$\frac{d(\text{GLF})}{d\mathbf{H}} = -\frac{\partial(\text{GLF})}{\partial\mathbf{u}} \left(J^{-1}\frac{\partial\mathbf{c}}{\partial\mathbf{H}}\right) + \frac{\partial(\text{GLF})}{\partial\mathbf{H}}. \tag{B4}$$

The main idea of the adjoint-based method is to introduce an auxiliary vector variable $\boldsymbol{\lambda}$ for solution of the *adjoint* system

$$J^T\boldsymbol{\lambda} = -\left(\frac{\partial(\text{GLF})}{\partial\mathbf{u}}\right)^T \tag{B5}$$

and then to compute the sensitivity as

$$\frac{d(\text{GLF})}{d\mathbf{H}} = \boldsymbol{\lambda}^T \frac{\partial\mathbf{c}}{\partial\mathbf{H}} + \frac{\partial(\text{GLF})}{\partial\mathbf{H}}. \tag{B6}$$

Equations (B4) and (B6) are equivalent, but the latter has the advantage of requiring the solution of a single linear system given by Equation (B5). In MALI, the Jacobian and the other derivatives, $\frac{\partial\mathbf{c}}{\partial\mathbf{H}}, \frac{\partial(\text{GLF})}{\partial\mathbf{u}}$, and $\frac{\partial(\text{GLF})}{\partial\mathbf{H}}$, are computed using automatic differentiation, a technique that allows for exact calculation of derivatives up to machine precision. For automatic-differentiation, MALI relies on the Trilinos *Sacado* package (Phipps and Pawlowski, 2012). As a final remark, we note that the term $\frac{\partial\mathbf{c}}{\partial\mathbf{H}}$ requires the computation of shape derivatives, because a change in thickness affects the geometry of the problem. This is not the case for two-dimensional, depth-integrated flow models (e.g., as in Goldberg et al. (2019)), or when using a sigma-coordinate to discretize the vertical dimension.



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
