# Peer review of "Diagnosing the sensitivity of grounding line flux to changes in sub-ice shelf melting"

_The Cryosphere, 2020_

## Referee Comment (RC1) · G. Hilmar Gudmundsson (Referee) · 15 Feb 2020

This is a very nice piece of work!

The authors present some important methodological advances. The use of the adjoint-method to calculate the grounding-line flux is very nice, and of course far better than the approach used in Reese et al (full disclosure, I was one of the authors of the paper, and it should have been my job to implement the adjoint approach myself for that work, but I was too busy with other things...So I'm, very glad that someone has now done what I myself should have done some time ago.)

I like the three research questions and I think the authors provide very satisfying answers to all of them in the paper. I wonder if the research question (1) could not be

formulated a bit better? Maybe: How do changes in ice-flux across grounding line, relate to estimates of ice-shelf buttressing evaluated at locations within the ice-shelf?

Must confess that I have never myself fully understood the usefulness of quantifying buttressing at some location within an ice shelf. What makes sense to me is to quantify the grounding-line buttressing provided by an ice shelf, and the changes in GL buttressing and GL ice flux as a function of thickness-perturbation across the ice shelf. I guess in some way the authors also address this issue when they conclude that the buttressing at a given location within an ice shelf depend critically on the normal direction chosen. (Possibly it might be better just to look at, in a general sense, how stresses within an ice shelf differ from the unconfined case, but again this will depend on the particular question being addressed.)

The authors investigate the possibly that perturbations in ground-line flux due to local changes in ice-shelf thickness, are linearly related to ice-shelf buttressing values calculated at those same locations within the ice shelf. This is an important point that needs to be investigated, and I guess it could be argued that Fuerst et al implicitly assumed this to be the case. (As mentioned above, I personally have never understood why one would expect there to be a simple correlation between these two quantities, except possibly in some general qualitative sense.) But this has been implicitly assumed in some previous work, and the authors are the first ones to actually look into this in any detail. They provide a detailed but arguably also a too long and somewhat confusing answer, but essentially I think they conclude that there is not simple relationship between these two. If I have correctly understood their conclusion, I would recommend stating more clearly this key finding and basically just write that there is no theoretical reason to expect GLF to scale in a simple way with buttressing numbers evaluated at a given location within the ice shelf, and that the numerical experiments show that no such simple relationship exists for the cases considered.

I had some difficulties following the discussion in 4.3.2. Not sure if this is really relevant, but a reduction in ice thickness change at any location within an ice shelf will generally

have two opposing effects on ice-shelf flow: 1) the spreading rate goes down and with it the speed further downstream 2) buttressing (as measured along the grounding lines upstream) will generally decrease and therefore speed increase. So there are two exactly opposing effects involved. Usually, reduction in ice-shelf thickness leads to an increase speed close to the grounding line, and decrease further downstream (provided the ice-shelf is long enough for the integrated effect of ice-shelf thinning to outweigh the effect on increased GL speed.)

The manuscript is still in a bit rough state. In fact, I find it to be in an unusually rough state compared to a typical TC submission. There are number of footnotes, and these seem to be mostly some comments aimed at the authors themselves. This needs to be sorted out and the presentation of the material needs to be sharpened up a bit (what are 'distal changes'?).

The figures are also some of rather poor quality. I guess most TC readers will know where the Larsen C is, but it might nevertheless be good to have some map showing the location of Larsen C.

It's a bit unusual to use curly brackets around a tensor as done in Eq, (11).

I missed an exact definition of the grounding-line flux and the GLF used in the adjoint method. Is it a line integral over all the grounding lines? How is the grounding-line defined at a local element level? Do you use the edges of the grounded elements, or do you cut through elements based on the flotation/grounding mask? If so, how do you interpolate velocities and ice thickness from the nodal points? Line 193-194: Not sure if I actually showed this. At least I don't think I used the concept of 'group velocity' in this context.

I would generally have recommended a minor revision to such an excellent work. But the presentation is still too poor, and for that reason I suggest a revision following a re-review.

---

## Referee Comment (RC2) · Anonymous Referee #2 · 12 Mar 2020

In this study, Zhang et al. present a detailed and thorough analysis of the relation between local ice-shelf buttressing numbers, how they are affected by local perturbations and how they relate to the flux response at the grounding line induced by such a local perturbation. They find that correlations between the flux response (see Reese et al., 2018) and the local buttressing numbers (see Fürst et al., 2016) can be found in very specific cases, but break down for more complicated geometries and when considering regions close to the grounding line. In a second step, they show how the adjoint method could be used to assess the sensitivity of grounding line flux to local perturbations which is shown to be consistent with the computationally more intense perturbation approach (except at the grounding line).

This study presents very interesting results that will help to advance the understanding of ice-shelf buttressing significantly. However, I think that some points should be addressed before it could be published.

Major comments

- Your manuscript would be much easier to read if the central questions and related main findings of your paper were formulated more clearly. This shows for example in your abstract, where you state that you search for causal connections between sub-shelf melting, buttressing and grounding line flux. However, there is no clear answer to that, you rather switch to presenting an alterative method to calculate the grounding line flux sensitivity in the second part of the abstract. This is also reflected, for example, in the formulation of the research questions on page 2, line 47-52.

- Section: 4.5: You state that the differences between the adjoint approach and the pertubation approach near the grounding line arise from 'nonlinearities' - more clarification is required at this point. In particular, Figures 13 and 14 show that the adjoint-sensitivites are negative along the grounding line, while Figure 15 indicates in general positive sensitivities in the perturbation approach. I think that the treatment of the grounding line in the sensitivity assessment could be key in explaining these differences. So please explain (1) how you specify the grounding line position in your experiments / model and how grounding line flux is calculated (can the grounding line move in your perturbation experiments?), (2) if these differences arise only for cells directly adjacent to the grounding line, and (3) how this is reflected in the adjoint method.
In addition, issues might arise due to the discretization. Perturbations in the ice shelf should theoretically not be able to change the ice thickness at the grounding line or the surface slope upstream, but they do so in numerical models, so it could be argued that including these regions is anyway problematic.

[Figure]

Further comments

- Title. In the manuscript you are not so much analysing the sensitivity of grounding line flux to perturbations itself, but you are rather (1) trying to relate the concept of buttressing numbers to the concept of locally-induced grounding line flux changes and (2) showing that the adjoint method is consistent with the pertubation approach. So I would suggest to reformulate your title to reflect the content better, maybe something in the direction of 'On the complicated relation between local ice-shelf buttressing and induced grounding line flux changes', 'Are there causal connections between local ice-shelf buttressing and locally-induced grounding line flux changes?' or 'Adjoint-based sensitivity of grounding line flux to sub-shelf melting...'

- page 1, line 7: I'm not sure if this is the correct argument to debunk a correlation between grounding line flux changes and local buttressing numbers.
  If the ice shelf is locally perturbed and buttressing at the grounding line is reduced, this speeds up ice flow at the grounding line up to the perturbation location. However, the perturbation will reduce the spreading rate and hence tends to reduce velocities at and downstream of the perturbation location. This shows in your figure 7 where velocities increase up to the perturbation location and decrease downstream of it, which is then reflected in a local reduction in longitudinal stresses. This is then interpreted as an increase in the local buttressing number based on, e.g., the flow direction. From this point of view, a reduction in buttressing at the grounding line can consistently be related to an increase in the local buttressing number.
  Your point here is supported by the fact that you cannot find correlations once you include regions close to the grounding line or you analyse Larsen C. Don't get me wrong, I think that it is a very important point to make that local perturbations increase locally measured buttressing numbers, as I do not think that many

people are aware of that (I was not).

- page 1, line 20 and other: please check your references, e.g., Schoof (2012) does not use idealized modelling and Asay-Davis et al. (2016) do not include experiments showing MISI, also Royston and Gudmundsson (2016) analyse diagnostic responses to ice-shelf collapse.

- page 2, line 43: 'diagnostic, forward experiments'

- page 4, line 86: you could add a subsection 'Initial configuration' here to improve readability.

- page 5, line 120: You need to multiply $N_{rp}$ with a time interval (e.g., one year if your flux is given in units per year) to get a dimensionless number.

- page 5, line 120: Do you exclude changes in grounding lines of ice rises in the Larsen C domain?

- page 5, line 121: you could add a subsection 'Local buttressing number' here to improve readability.

- Figure 2: labels for the colorbars are missing and it would be easier to understand your message if you added the normal directions in the panels (also in Figure 3 and others).

- page 7, line 156: Why $12$km? Does the relation already improve if you remove only cells that are directly linked to the grounding line?

- Figure 4: Please add p-values for your correlation statistics.

- page 8, line 174, isn't this a contradiction to your statement in p7., line 58?

- page 11, line 195, maybe better state that the thickness gradients magnitude increases / decreases, since this is the relevant quantity to drive ice flow.

- page 11, line 204: I do not understand your sentence in brackets, please clarify.

- Figure 8: why do you analyse buttressing changes in neighboring cells and not in the cell itself? This should not make a difference, given Fig. 7 etc, or do I miss some argument here?

- page 12, line 218: you could add that this negative correlation is in line with the general understanding of how buttressing reduction affects ice flow.

- Section 4.3.2.: when calculating buttressing at the grounding line, you have an additional direction that emerges naturally, which is the grounding line normal as used in Gudmundsson (2013). In fact, since the boundary condition at the calving front is formulated in terms of the calving front normal, this is the only direction that guarantees that you get a value of $1$ if and only if you do not have any buttressing at the corresponding grounding line location. It is worth checking, how using that normal affects your findings.

- page 13, line 235: I do not understand your statement here as there is a difference between the first principal component along the grounding line and within the ice shelf?

- page 14, line 245: you state that you test experiments with and without perturbing elements that are crossed by the grounding line, but you never refer to these experiments again.

- page 15, line 266: It might be worth checking the flow and normal directions as well (similar to Figure S8 for the p1-direction).

- page 17, line 300: I suppose that you repeat the perturbations for the different thicknesses?

- page 17, line 314: you refer here to the other methods discussed in the previous sections, i.e., the local buttressing numbers etc?

- page 17, footnote: Are you comparing here with different perturbation experiments than those presented in the sections before? Please clarify.

- Figures could be improved substantially by adding labels and units, making sure that font size allows for readability etc.

- page 20, line 332: In addition, the analysis by Fürst et al. (2016) might be based on 'maximum buttressing' since the second principal stress is related to the notion of the compressive arch.

- page 23, line 401: this could be misundertsood ('ice thickness vector'), maybe easier if you specify $(H_n)_{n\in nodes}$ and $(u_n, v_n)_{n\in nodes}$ or something like this? And shouldn't the grounding line flux depend on the velocities (not just their magnitude/speed)?

- page 23, line 404: please specify $i$ and $j$

**References**

Asay-Davis, X. S., Cornford, S. L., Durand, G., Galton-Fenzi, B. K., Gladstone, R. M., Gudmundsson, G. H., Hattermann, T., Holland, D. M., Holland, D., Holland, P. R., et al. (2016). Experimental design for three interrelated marine ice sheet and ocean model intercomparison projects: Mismip v. 3 (mismip+), isomip v. 2 (isomip+) and misomip v. 1 (misomip1). *Geoscientific Model Development*, 9(7):2471–2497.

Fürst, J. J., Durand, G., Gillet-Chaulet, F., Tavard, L., Rankl, M., Braun, M., and Gagliardini, O. (2016). The safety band of antarctic ice shelves. *Nature Climate Change*, 6(5):479.

Gudmundsson, H. (2013). Ice-shelf buttressing and the stability of marine ice sheets. *The Cryosphere*, 7(2):647–655.

Reese, R., Gudmundsson, G. H., Levermann, A., and Winkelmann, R. (2018). The far reach of ice-shelf thinning in antarctica. *Nature Climate Change*, 8(1):53–57.

Royston, S. and Gudmundsson, G. H. (2016). Changes in ice-shelf buttressing following the collapse of larsen a ice shelf, antarctica, and the resulting impact on tributaries. *Journal of Glaciology*, 62(235):905–911.

Schoof, C. (2007). Ice sheet grounding line dynamics: Steady states, stability, and hysteresis. *Journal of Geophysical Research: Earth Surface*, 112(F3).

Schoof, C. (2012). Marine ice sheet stability. *Journal of Fluid Mechanics*, 698:62–72.

---

## Referee Comment (RC3) · Anonymous Referee #3 · 13 Mar 2020

Summary and comments on the manuscript entitled
**Diagnosing the sensitivity of grounding line flux to changes in sub-ice shelf melting**
presented on 11.02.2020

by

T. Zhang et al.

**Summary**

In this manuscript, the author employ a new state-of-the-art ice-flow model and assess the utility of various buttressing metrics for inferring grounding line response to distant ice-shelf thinning. For this purpose, they reconcile two former studies that introduce metrics for the local ice-shelf buttressing and the integrated flux response along the grounding line (GL). For local thinning perturbations, the authors show that for relevant parts of an ice shelf (away from the GL and unconfined parts of the ice-shelf), there is a positive correlation between the two metrics. Highest values are found when the buttressing metrics is computed along the first principal stress direction (p1). Yet, buttressing values increase in the vicinity of the thinning perturbations, which seems counter-intuitive with respect to the concurrent increase in the grounding line flux (GLF). This finding makes changes in ice-shelf buttressing utterly difficult to interpret. In a final step, an adjoint-based GLF sensitivity is computed, which shows comparability to results from a large ensemble of forward evaluations. This sensitivity measure has the potential to be very useful in delineating ice-shelf areas relevant for restraining present outlet-glacier discharge.

I gladly admit that I was very excited about this study because the authors present a computationally efficient adjoint-based method to compute GLF sensitivities that gives identical results as a cumbersome diagnostic perturbation ensemble. Initially, they also convinced me about the limited utility of changes in the buttressing index. Yet after plunging into the review, I strongly contest this judgment because the underlying analysis seems somewhat biased (see below) and I urge the authors to moderate their assessment. The authors themselves show that the buttressing index along the p1-direction is actually very informative in terms of GLF sensitivity. This is a very important conclusion, which will be appreciated by modellers that cannot compute this adjoint-based sensitivity. Moreover, I have identified a potential error in the index calculation, which might have severe implications.

In summary, I remain very positive about this manuscript and I recommend that the editor should continue to considered it for publication in *The Cryosphere* after my concerns have been alleviated. This will require a major revision during which a fundamental change in the manuscript structure might be necessary.

**General comments**

**Erroneous calculation**

From a vertically integrated perspective, the normal stress $T_{nn}$ which is computed in the various directions should be maximal and minimal for the first (p1) and second (p2) principal stress directions, respectively. This implies that the buttressing is minimal in p1 and maximal in p2 direction (you show this nicely in Figure S1 yourself). In Figure 2, you show the buttressing values for the MISMIP+ setup in various directions. While the p2-values appear maximal, the p1 values seem larger than the values computed in flow direction. This cannot be correct. I suspect that you confused panels b) and c). If not, this comment might have severe implications. Please verify.

**Inconsistent and biased analysis**

I certainly appreciate how carefully you have structured the analysis in this manuscript. You clearly state a correlation between the GLF response and the buttressing index in dynamically relevant areas (cf., Sect. 4.2, Fig.4). Thereafter, you show that buttressing changes in the vicinity of the thinning perturbations exhibit a counter-intuitive behaviour which is difficult to interpret. Yet, this difficulty seems to have entirely undermined your confidence in the interpretability of this measure. In the abstract, you even condemn the correlation between the GLF and the local buttressing measures as remaining '[...] elusive from a physical perspective'. This judgment is evoked throughout your manuscript and I somehow feel that I have to take up the cudgels for this metric. First, you show yourself that there can be good correlation (Figs. 4,5,11b). The more I tried to understand the details of your analysis, I have more and more doubts about its robustness. First doubts arose when I read through Sect. 4.3.1. You start by discusseing non-local speed-up in the vicinity of the perturbed area (but excluding the centre). Thereafter, you focus on the local-scale buttressing changes within the perturbed area. This seemed inconsistent and this choice biases and discredits the buttressing change measure. Initially, I was willing to accept this judgment but then I realised that the same counter-intuitive response is seen in the principal strain-rate components (Fig.7$e$ and $f$). These also indicate compression within the perturbed area (and slightly beyond). Consequently, you also need to dismiss the usefulness of this measure. This is too much of a stretch for me. I simply think that your analysis should consistently avoid areas close to the perturbations. To substantiate my view, I want to briefly explain the 1st principal buttressing or strain-rate changes in Fig7 $e$ and $g$. After the perturbation, you clearly get less buttressing and increased extension upstream and downstream (in-flow direction) of the affected area. Sideways, but still along the 1st principal direction, these effects result in increased buttressing and compression (similar to a bottleneck effect). This explanation seems reasonable. I therefore strongly urge you to moderate and adjust your assessment of the buttressing metric, accordingly.

A main motivation for why I raise this point is that many ice-flow models are not capable of an adjoint-based evaluation. It would therefore be constructive, if you could give some advice on how to best evaluate the local buttressing metric wrt. the GLF sensitivity. You nicely show that there is a correlation. Your strategy to introduce a buffer zone around the grounding line is valuable (it is anyhow clear that these regions are important for the GLF sensitivity). From Fig.4, I think that areas with negative buttressing values should also be excluded (gives more flexibility than prescribed masking). So you could give some advise on how this metric can still be useful. Moreover, you should emphasise that if the interest is in the GLF sensitivity, the buttressing metric should be computed in p1/flow direction as against Fürst et al. (2016). This comment further implies that you might want to reconsider the structure of the document: I suggest that you start with the GLF sensitivity of Reese et al. (2018). Then you could show that the adjoint-based approach gives equivalent results. Afterwards, you might want to asses the utility of the buttressing metrics (advice, limitations, etc.) to explain the GFL sensitivity.

**Minimum and maximum speed increase**
It took me a while to get my head around the retrieval of the direction of the minimal and maximal speed increase (L182ff). Although I am very impressed by the distinct peaks in the resultant distribution (Fig.6b), I wonder about its utility in this study. After its presentation, this measure is briefly compared to Gudmundson (2003) and shortly re-raised for the Larsen C setup. It is not discussed nor mentioned in the conclusions. I therefore urge you to re-consider its utility.

**Specific comments**

1. Please reduce the overall amount of footnotes. Sometimes they keep valuable extra information, which should appear in the text.

2. Please introduce a figure of GLF response $N_{\mathrm{rp}}$ and the buttressing values (p1,p2, flow) for Larsen C. It might help you to delineate the area in which the GLF response and buttressing values are correlated.

**Detailed comments**

**L29** The term 'longitudinal stresses' seem to be too narrow here. I would rather speak of 'membrane stresses' following Hindmarsh (2006).
**L42** Delete 'of ice'

**L60** Insert comma after parenthesis.

**L115** This sentence is not true. You do not show the response on the southern/botton part of the MISMIP+ setup.

**L118** As in the original study by Reese et al. (2018), I do not understand the meaning of $P$. You say it is the local mass change associated with the perturbation. So it should be rather constant (despite element size variations on Larsen C). Units should be $m^3$. The GL flux change $R$ is however in units of $m^3/yr$. I do not understand how $N_{\text{rp}}$ can then be dimensionless. I think that I misunderstood the meaning of $P$. Please explain in more detail.

**L169** You must have noticed the dip in the correlation with the p1-buttressing (Fig. 5a). So the best correlation occurs at $\pm 25°$. With respect to the flow direction, the optimal correlation is ~$100°$ turned (counterclockwise, Fig. 5b). Your statement in this line does note entirely hold.

**L173** You envoke the notion of an overall best buttressing metric. I do not think that this exists as such. It will depend on the spatial focus which can be the GL, central areas of the ice shelf or the calving front. Please remove this notion of a best metric.

**L194-L207** Prior to this section, you focus on the speed-up signal 'among neighbouring cells' (L182-L194). In this section, you then discuss buttressing changes within the perturbed areas. This seems inconsistent. From Fig. 7$g$ and $h$, I think you can extract a meaningful, aggregated index for buttressing changes excluding the perturbation centre. Upstream of the perturbation (in flow or p1 direction), the buttressing decreases with highest decrease close to the perturbed area. This inconsistent treatment therefore seems deliberate and strongly biases your interpretation. This bias leads to harsh judgments of the buttressing metric in the subsequent two sections, which are, in my opinion, note well justified. Please stay more objective. You also show the strain rates fields in the principal direction which also show overall compression within the perturbed zones. You do no discredit the usefulness of these values either.

**L225-L238** This paragraph judges the results and it is therefore better located in the discussion conclusion. I also sense some redundancy.

**L273-L289** This paragraph presents methodology so it should appear earlier (not as a sub-section of the Results).

**Fig.1** Poor figure quality. Missing overview figure for localisation of Larsen C. What did you do about Bawden Ice Rise? Could you also show the observed velocity magnitude on Larsen C. Please indicate in the figure that the velocities you show, present the state after the relaxation (you only mention this in the text L105).

**Fig.2** In the caption you speak about 'perturbation points'. The perturbation does not affect a single point but an entire patch. I would use different colours for the response number and buttressing metrics. Why do you get negative response numbers for perturbations next to the grounding line?

**Figs.11&12** I would try to merge these figures. Panels (a) can be placed as an inset into panels (b).

**References**

Fürst, J., Durand, G., Gillet-Chaulet, F., Tavard, L., Rankl, M., Braun, M., and Gagliardini, O.: The safety band of Antarctic ice shelves, Nature Climate Change, 6, 479–482, doi:10.1038/nclimate2912, 2016.

Hindmarsh, R.: The role of membrane-like stresses in determining the stability and sensitivity of the Antarctic ice sheets: back pressure and grounding line motion, Philosophical Transactions of the Royal Society A, 364, 1733–1767, doi:10.1098/rsta.2006.1797, 2006.

Reese, R., Gudmundsson, G., Levermann, A., and Winkelmann, R.: The far reach of ice-shelf thinning in Antarctica, The Cryosphere, 8, 53–57, doi: 10.1038/s41558-017-0020-x, 2018.

---

## Author Comment (AC1) · 7 Jun 2020

**Review 1 (reviewer comments in italic)**

*This is a very nice piece of work!*
*The authors present some important methodological advances. The use of the adjoint method to calculate the grounding-line flux is very nice, and of course far better than the approach used in Reese et al (full disclosure, I was one of the authors of the paper, and it should have been my job to implement the adjoint approach myself for that work, but I was too busy with other things. . .So I'm very glad that someone has now done what I myself should have done some time ago.)*

We appreciate the general endorsement of the work and the helpful suggestions below, which we have used to clarify and improve the paper.

*I like the three research questions and I think the authors provide very satisfying answers to all of them in the paper. I wonder if the research question (1) could not be formulated a bit better? Maybe: How do changes in ice-flux across grounding line relate to estimates of ice-shelf buttressing evaluated at locations within the ice-shelf?*

Thanks for the suggestion. We have changed the initial version of this question "Do local evaluations of ice shelf buttressing reflect how local perturbations in ice shelf thickness impact grounding line flux?" to "How do changes in ice-flux across the grounding line relate to local estimates of ice-shelf buttressing evaluated on the ice-shelf?"

*Must confess that I have never myself fully understood the usefulness of quantifying buttressing at some location within an ice shelf. What makes sense to me is to quantify the grounding-line buttressing provided by an ice shelf, and the changes in GL buttressing and GL ice flux as a function of thickness-perturbation across the ice shelf. I guess in some way the authors also address this issue when they conclude that the buttressing at a given location within an ice shelf depend critically on the normal direction chosen. (Possibly it might be better just to look at, in a general sense, how stresses within an ice shelf differ from the unconfined case, but again this will depend on the particular question being addressed.)*

*The authors investigate the possibly that perturbations in ground-line flux due to local changes in ice-shelf thickness, are linearly related to ice-shelf buttressing values calculated at those same locations within the ice shelf. This is an important point that needs to be investigated, and I guess it could be argued that Fuerst et al implicitly assumed this to be the case. (As mentioned above, I personally have never understood why one would expect there to be a simple correlation between these two quantities, except possibly in some general qualitative sense.) But this has been implicitly assumed in some previous work, and the authors are the first ones to actually look into this in any detail. They provide a detailed but arguably also a too long and somewhat confusing answer, but essentially I think they conclude that there is not simple relationship between these two. If I have correctly understood their conclusion, I would recommend stating more clearly this key finding and basically just write that there is no*

*theoretical reason to expect GLF to scale in a simple way with buttressing numbers evaluated at a given location within the ice shelf, and that the numerical experiments show that no such simple relationship exists for the cases considered.*

As suggested by the reviewer, throughout the revised manuscript we have attempted to (1) clarify that this is one of our primary conclusions, (2) more clearly tie together the various sections of the paper that need to be understood to support this conclusion, and (3) justify this conclusion based on the results and discussion presented in the paper.

*I had some difficulties following the discussion in 4.3.2. Not sure if this is really relevant, but a reduction in ice thickness change at any location within an ice shelf will generally have two opposing effects on ice-shelf flow: 1) the spreading rate goes down and with it the speed further downstream 2) buttressing (as measured along the grounding lines upstream) will generally decrease and therefore speed increase. So there are two exactly opposing effects involved. Usually, reduction in ice-shelf thickness leads to an increase speed close to the grounding line, and decrease further downstream (provided the ice-shelf is long enough for the integrated effect of ice-shelf thinning to outweigh the effect on increased GL speed.)*

*The manuscript is still in a bit rough state. In fact, I find it to be in an unusually rough state compared to a typical TC submission. There are number of footnotes, and these seem to be mostly some comments aimed at the authors themselves.*

*This needs to be sorted out and the presentation of the material needs to be sharpened up a bit (what are 'distal changes'?).*

We have largely rewritten all of section 4.3 and attempted to clarify the main points therein and better connect them to the preceding and following sections, in order to better support the main findings of the paper. We also have removed most of the footnotes, either deleting them or incorporating them directly into the main text.

*The figures are also some of rather poor quality. I guess most TC readers will know where the Larsen C is, but it might nevertheless be good to have some map showing the location of Larsen C.*

A location figure for Larsen C has been added to Figure 1. We have also added several other new figures (mainly to the SI, in response to other reviewers). In general, the majority of figures and their captions have been updated and improved.

*It's a bit unusual to use curly brackets around a tensor as done in Eq, (11).*

The brackets in Eq. (11) have been changed to parentheses.

*I missed an exact definition of the grounding-line flux and the GLF used in the adjoint method. Is it a line integral over all the grounding lines?*

The reviewer is correct. We have added a new section up front (2.1.1) to clarify how the grounding line flux is calculated (along with additional discussion relating to the adjoint approach in Appendix C).

*How is the grounding-line defined at a local element level? Do you use the edges of the grounded elements, or do you cut through elements based on the flotation/grounding mask? If so, how do you interpolate velocities and ice thickness from the nodal points?*

The location of the grounding line in the model is defined at sub-grid resolution (i.e., "cut through") using a floating vs. grounded mask. Velocities and thickness are discretized as nodal finite element fields and are evaluated on the grounding line. This is now explained in detail in the new section 2.1.1.

*Line 193-194: Not sure if I actually showed this. At least I don't think I used the concept of 'group velocity' in this context.*

We have removed this discussion in the updated version of the manuscript so this is no longer relevant.

*I would generally have recommended a minor revision to such an excellent work. But the presentation is still too poor, and for that reason I suggest a revision following a re-review.*

The manuscript has been significantly revised since the initial submission and we believe that all of the reviewers concerns are adequately addressed.

---

## Author Comment (AC2) · 7 Jun 2020

**Review 2 (reviewer comments in italic)**

*In this study, Zhang et al. present a detailed and thorough analysis of the relation between local ice-shelf buttressing numbers, how they are affected by local perturbations and how they relate to the flux response at the grounding line induced by such a local perturbation. They find that correlations between the flux response (see Reese et al., 2018) and the local buttressing numbers (see Fürst et al., 2016) can be found in very specific cases, but break down for more complicated geometries and when considering regions close to the grounding line. In a second step, they show how the adjoint method could be used to assess the sensitivity of grounding line flux to local perturbations which is shown to be consistent with the computationally more intense perturbation approach (except at the grounding line).*

*This study presents very interesting results that will help to advance the understanding of ice-shelf buttressing significantly. However, I think that some points should be addressed before it could be published.*

We thank the reviewer for all of the helpful comments below.

*Major comments*

*Your manuscript would be much easier to read if the central questions and related main findings of your paper were formulated more clearly. This shows for example in your abstract, where you state that you search for causal connections between sub-shelf melting, buttressing and grounding line flux. However, there is no clear answer to that, you rather switch to presenting an alterative method to calculate the grounding line flux sensitivity in the second part of the abstract. This is also reflected, for example, in the formulation of the research questions on page 2, line 47-52.*

We have modified the abstract to remove the focus on understanding the causal connections between melt perturbations, buttressing, and grounding line flux. The research questions in the introduction have also been slightly modified (also in response to reviewer no. 1). As discussed further below, we have also significantly updated and improved the content and writing relative to our initial submission.

*Section: 4.5: You state that the differences between the adjoint approach and the perturbation approach near the grounding line arise from 'nonlinearities' - more clarification is required at this point. In particular, Figures 13 and 14 show that the adjoint-sensitivites are negative along the grounding line, while Figure 15 indicates in general positive sensitivities in the perturbation approach. I think that the treatment of the grounding line in the sensitivity assessment could be key in explaining these differences.*

We interpret the large nonlinearities close to the grounding line as a consequence of 1) relatively large thickness gradients and the rapid transition from vertical-shear-dominated to

membrane-stress-dominated flow close to the grounding line, and 2) the changes of grounding line position and local geometry due to the change of thickness in the cells adjacent to the grounding line. This is now more clearly stated in the manuscript.

In terms of the difference between Figs. 13, 14 versus Fig. 15 (11, 12, and 13 in the revised version), the analysis in current Fig. 13 focuses on a small fraction of the total number of perturbation points -- those near the grounding line -- whereas current Figs. 11, 12 include those same points plus many more points (all in the case of MISMIP+) on the ice shelf proper. Thus, there are relatively more negative values obvious in current Fig. 13.

*So please explain (1) how you specify the grounding line position in your experiments / model and how grounding line flux is calculated (can the grounding line move in your perturbation experiments?),*

We have added a new section, 2.1.1, in which we detail the computation of the GL and the GLF. We also note that a perturbation to the ice thickness at a (triangular) grid cell intersecting the GL will affect both the GL position and length (because the thickness affects the flotation condition, which in turn can affect the position of the GL).

*(2) if these differences arise only for cells directly adjacent to the grounding line, and*

We observe negative sensitivities only for cells intersecting the grounding line. This can also happen for the perturbation-based sensitivity approach, as shown in Figure 2a (for the Larsen C case, the perturbation points are randomly chosen and we are not reporting results for all of the points at the grounding line). Differences between the adjoint-based and the perturbation-based sensitivities are more pronounced near the grounding line and become smaller far from the grounding line, as shown in revised Figure 13 of the paper.

*(3) how this is reflected in the adjoint method. In addition, issues might arise due to the discretization. Perturbations in the ice shelf should theoretically not be able to change the ice thickness at the grounding line or the surface slope upstream, but they do so in numerical models, so it could be argued that including these regions is anyway problematic*

The reviewer is correct that perturbations on the ice shelf proper do not change the thickness or slope of the ice at the grounding line while they do in the case where a grid cell being perturbed also intersects with the grounding line (as now explained more carefully in section 2.1.1). For this reason we argue that one should refrain from including cells that intersect the grounding line when performing similar sensitivity analysis (when using either the adjoint- or perturbation-based approach). Further, one should also refine the mesh near the GL to get more accurate sensitivities near the GL. This is now stressed more clearly in section 4.5.

The adjoint method also accounts for possible changes in the GL position/length due to (infinitesimal) changes in the ice thickness at triangular grid cells intersecting the GL. This is now explained in more detail in (new) section 2.1.2 and Appendix C after equation C1.

We further note that, in the numerical model, the thickness on the ice shelf and along the grounding line does not change *prognostically.*

*Title: In the manuscript you are not so much analysing the sensitivity of grounding line flux to perturbations itself, but you are rather (1) trying to relate the concept of buttressing numbers to the concept of locally-induced grounding line flux changes and (2) showing that the adjoint method is consistent with the perturbation approach. So I would suggest to reformulate your title to reflect the content better, maybe something in the direction of 'On the complicated relation between local ice-shelf buttressing and induced grounding line flux changes', 'Are there causal connections between local ice-shelf buttressing and locally-induced grounding line flux changes?' or 'Adjoint-based sensitivity of grounding line flux to sub-shelf melting...'*

We appreciate the reviewer's suggestion. However, we have decided to keep the original title as we feel that it clearly encompasses and describes our efforts and the main focus of the paper.

*page 1, line 7: I'm not sure if this is the correct argument to debunk a correlation between grounding line flux changes and local buttressing numbers. If the ice shelf is locally perturbed and buttressing at the grounding line is reduced, this speeds up ice flow at the grounding line up to the perturbation location. However, the perturbation will reduce the spreading rate and hence tends to reduce velocities at and downstream of the perturbation location. This shows in your figure 7 where velocities increase up to the perturbation location and decrease downstream of it, which is then reflected in a local reduction in longitudinal stresses. This is then interpreted as an increase in the local buttressing number based on, e.g., the flow direction. From this point of view, a reduction in buttressing at the grounding line can consistently be related to an increase in the local buttressing number. Your point here is supported by the fact that you cannot find correlations once you include regions close to the grounding line or you analyse Larsen C. Don't get me wrong, I think that it is a very important point to make that local perturbations increase locally measured buttressing numbers, as I do not think that many people are aware of that (I was not).*

In general, the reviewer seems to be largely agreeing with the interpretation and conclusions we present in our paper. That is, that a local *increase* in buttressing number following perturbations is paradoxical with respect to the overall *decrease* in buttressing experienced by the broader ice shelf (leading to an *increase* in ice flux across the grounding line). Further, as we discuss in more detail in our revised manuscript (specifically, in sections 4.3.1 and 4.3.2), the local (at the perturbation location) change in buttressing is not always consistent and is often in opposition to the broader response immediately neighboring it. The reviewer may be proposing that the increase in buttressing number in the local area around a perturbation is a consistent diagnostic that could be used as a proxy for overall increases in grounding line flux. While this may be the

case for specific domains and perturbations, we are not comfortable making such a general statement here based on the analysis conducted.

*page 1, line 20 and other: please check your references, e.g., Schoof (2012) does not use idealized modelling and Asay-Davis et al. (2016) do not include experiments showing MISI, also Royston and Gudmundsson (2016) analyse diagnostic responses to ice-shelf collapse.*

We have changed the Schoof (2012) reference to the more appropriate Schoof (2007) reference. The Asay-Davis et al. (2016) reference was meant to be a placeholder for the MISMIP+ paper (which at the time of this submission, had not yet been submitted). We have updated it to the correct reference for the MISMIP+ experiments (Cornford et al. 2020).

*page 2, line 43: 'diagnostic, forward experiments'*

Changed.

*page 4, line 86: you could add a subsection 'Initial configuration' here to improve readability.*

Changed as suggested. A new "Model configuration" subsection (2.2) has been added.

*page 5, line 120: You need to multiply Nrp with a time interval (e.g., one year if your flux is given in units per year) to get a dimensionless number.*

In Reese et al. (2018), it is implicit that the time period of interest is one year (according to personal communications). Therefore, P should have units of $m^3$, which are the same units as R. We have explicitly stated that the units of R and P are both in $m^3$ (in which case their ratio is non-dimensional).

*page 5, line 120: Do you exclude changes in grounding lines of ice rises in the Larsen C domain?*

For the Larsen C domain, the grounding lines around ice rises are treated in the same way as the "primary" grounding line. Therefore, the cells close to ice rises will also be removed when we pick cells for analyzing based on the distance to the grounding line. We have added an explicit statement about the treatment of ice rises to section 2.1.1.

*page 5, line 121: you could add a subsection 'Local buttressing number' here to improve readability.*

Since this section is already fairly short, and to avoid breaking up the paper into too many short sections, we've opted to keep the discussion of the local buttressing number in with the discussion of the flux response number (and the perturbation experiments in general).

*Figure 2: labels for the colorbars are missing and it would be easier to understand your message if you added the normal directions in the panels (also in Figure 3 and others).*

We have added colorbar titles and also labels for the choice of buttressing-number normal direction to Figs. 2, 3 and 4.

*page 7, line 156: Why 12km? Does the relation already improve if you remove only cells that are directly linked to the grounding line?*

We have substantially revised this portion of the paper because we discovered a new metric that can be used for "removing" these areas from consideration (based largely on whether or not the region is experiencing significant shear and/or close to the grounding line). This is discussed in the update section 4.1 (and new Figure S2).

*Figure 4: Please add p-values for your correlation statistics.*

The p values, relative to the null hypothesis that $N_{rp}$ is independent of $N_b$, are 1.23e-59 for $\mathbf{n}_{p1}$, 1.57e-09 for $\mathbf{n}_{p2}$ and 5.20e-31 for $\mathbf{n}_{flow}$. However, we think that it would be misleading to report these exceedingly small p-values in the paper. Here, we are trying to assess whether $N_{rp}$ and $N_b$ are linearly related and for this reason we are using linear regression to compute the fitting line and the correlation coefficient, to quantify the discrepancy from that line. We are not interested in ascribing a statistical interpretation to this line fit (which would then need to be explained / defended). For these reasons, we argue against including p-values in the paper.

*page 8, line 174, isn't this a contradiction to your statement in p7., line 58?*

We have changed Line 158 (now Line 198; end of section 4.2) to "... we find that buttressing in this direction is not useful for predicting changes in GLF; compared to $N_b(\mathbf{n}_{p2})$, $N_b(\mathbf{n}_{p1})$ and $N_b(\mathbf{n}_f)$ both show a better correlation with changes in GLF via local, sub-ice shelf melt perturbations.", for consistency with old line 174.

*page 11, line 195, maybe better state that the thickness gradients magnitude increases / decreases, since this is the relevant quantity to drive ice flow.*

Because the change in thickness gradient *magnitude* doesn't allow for any information regarding how the change in ice thickness impacts the *direction* of the ice flow, we prefer to keep the wording here as is. This is important because with no information regarding the sign of the gradient change, it's not immediately obvious if the change in thickness gradient should lead to an increase or a decrease in the local ice velocity.

*page 11, line 204: I do not understand your sentence in brackets, please clarify.*

This section has been completely re-written.

*Figure 8: why do you analyse buttressing changes in neighboring cells and not in the cell itself? This should not make a difference, given Fig. 7 etc, or do I miss some argument here?*

In 4.3.1 and 4.3.2 we now discuss in detail the buttressing changes both at the perturbation location and in the immediately surrounding neighborhood.

*page 12, line 218: you could add that this negative correlation is in line with the general understanding of how buttressing reduction affects ice flow.*

We have revised this sentence to state this more explicitly (around line 266 in the revised manuscript).

*Section 4.3.2.: when calculating buttressing at the grounding line, you have an additional direction that emerges naturally, which is the grounding line normal as used in Gudmundsson (2013). In fact, since the boundary condition at the calving front is formulated in terms of the calving front normal, this is the only direction that guarantees that you get a value of 1 if and only if you do not have any buttressing at the corresponding grounding line location. It is worth checking, how using that normal affects your findings.*

We have updated this figure and the discussion around it, including (as suggested) the addition of a subplot showing the value of Eq. 14 when changes in buttressing are calculated in the direction normal to the grounding line (of Fig. 8a).

*page 13, line 235: I do not understand your statement here as there is a difference between the first principal component along the grounding line and within the ice shelf?*

This statement has been removed.

*page 14, line 245: you state that you test experiments with and without perturbing elements that are crossed by the grounding line, but you never refer to these experiments again.*

This section has been entirely revised. Updated versions of the figures (and the related discussion) from the original version are now included in the SI. With respect to this comment, we do not include cells that cross the grounding line but rather grid cells that are *close to the grounding line* but still on the ice shelf. These experiments are discussed in more detail in the following two paragraphs. Note that we have also edited this sentence to try to clarify its meaning.

*page 15, line 266: It might be worth checking the flow and normal directions as well (similar to Figure S8 for the p1-direction).*

The relevant figures are now all contained by Fig. S6 and the related discussion is also in the SI. Note that in those figures, we show correlations for all directions (i.e., 180 degrees around the p1 direction).

*page 17, line 300: I suppose that you repeat the perturbations for the different thicknesses?*

Correct. The experiments are repeated with different sized thickness perturbations. For clarity, we've added "the only change being the magnitude of the applied perturbation" in Line 329.

*page 17, line 314: you refer here to the other methods discussed in the previous sections, i.e., the local buttressing numbers etc?*

Correct. We've added a clarifying statement "Two previous approaches for assessing GLF sensitivity to changes in ice shelf buttressing – the flux response number ($N_{rp}$) and the buttressing number ($N_b$)..." to this effect (line 357).

---

## Author Comment (AC3) · 7 Jun 2020

**Review 3 (reviewer comments in italic)**

*Summary*

*In this manuscript, the author employ a new state-of-the-art ice-flow model and assess the utility of various buttressing metrics for inferring grounding line response to distant ice-shelf thinning. For this purpose, they reconcile two former studies that introduce metrics for the local ice-shelf buttressing and the integrated flux response along the grounding line (GL). For local thinning perturbations, the authors show that for relevant parts of an ice shelf (away from the GL and unconfined parts of the ice-shelf), there is a positive correlation between the two metrics. Highest values are found when the buttressing metrics is computed along the first principal stress direction (p1). Yet, buttressing values increase in the vicinity of the thinning perturbations, which seems counter-intuitive with respect to the concurrent increase in the grounding line flux (GLF). This finding makes changes in ice-shelf buttressing utterly difficult to interpret. In a final step, an adjoint-based GLF sensitivity is computed, which shows comparability to results from a large ensemble of forward evaluations. This sensitivity measure has the potential to be very useful in delineating ice-shelf areas relevant for restraining present outlet-glacier discharge.*

In order to avoid possible misunderstandings, we point out that we do not conduct any forward model (i.e., prognostic) evaluations here. All experiments are strictly diagnostic in nature. We have stated this clearly in the revised manuscript.

*I gladly admit that I was very excited about this study because the authors present a computationally efficient adjoint-based method to compute GLF sensitivities that gives identical results as a cumbersome diagnostic perturbation ensemble. Initially, they also convinced me about the limited utility of changes in the buttressing index. Yet after plunging into the review, I strongly contest this judgment because the underlying analysis seems somewhat biased (see below) and I urge the authors to moderate their assessment. The authors themselves show that the buttressing index along the p1-direction is actually very informative in terms of GLF sensitivity. This is a very important conclusion, which will be appreciated by modellers that cannot compute this adjoint-based sensitivity. Moreover, I have identified a potential error in the index calculation, which might have severe implications.*

We were also initially similarly excited by the possibility of a computationally inexpensive and easily calculated metric for use in assessing the impact of local ice shelf thickness changes on changes in grounding line flux (i.e., a way to obtain the information from the Reese et al. calculations but with less effort). Indeed, this was an initial goal of our research, along with providing some physical basis for better understanding and justifying the apparent correlations between local measures of ice shelf buttressing and changes in grounding line flux.

In the end, however, we concluded that we cannot in good faith make a recommendation for using these apparent correlations. First, we've found it difficult to provide a clear explanation for their existence (i.e., the physical mechanisms connecting them). Second, we've found and

demonstrated clear contradictions between changes in buttressing on the shelf, in the vicinity of perturbations, and changes in integrated buttressing and grounding line flux, which are contrary to our understanding for how ice shelf buttressing works. Most important, however, is that even for simple or idealized ice shelf geometries, numerous data points near the grounding line -- the region that is most sensitive to perturbations -- must be removed for strong correlations to emerge. For a realistic ice shelf, only a small number of points near the center of the ice shelf remain useful at demonstrating the correlation. Lastly, as we show in a newly added Supplemental Table, there are many other physical quantities that correlate with changes in grounding line flux, some of which may simply be fortuitous or spurious (and, as with the buttressing number, we find that these same correlations become much weaker and less convincing when applied to realistic domains). Thus, while we appreciate the reviewer's comments, we argue that it is not within the goals or scope of the current work to come up with additional reasons to further justify the application of these easy-to-calculate metrics.

The potential error the reviewer alludes to is the swap of panels b and d in Figure 2. This is, however, an isolated mistake with no implications regarding the analysis conducted in the rest of the paper (as noted further below).

*In summary, I remain very positive about this manuscript and I recommend that the editor should continue to consider it for publication in The Cryosphere after my concerns have been alleviated. This will require a major revision during which a fundamental change in the manuscript structure might be necessary.*

We appreciate the reviewer's careful attention to our manuscript and, as detailed below, address as many of their concerns as we can without changing the fundamental interpretation of our results. We also note that many sections of the paper have been significantly revised relative to the initial submission, including additional analysis, arguments, and changes in presentation.

*Erroneous calculation: From a vertically integrated perspective, the normal stress Tnn which is computed in the various directions should be maximal and minimal for the first (p1) and second (p2) principal stress directions, respectively. This implies that the buttressing is minimal in p1 and maximal in p2 direction (you show this nicely in Figure S1 yourself). In Figure 2, you show the buttressing values for the MISMIP+ setup in various directions. While the p2-values appear maximal, the p1 values seem larger than the values computed in flow direction. This cannot be correct. I suspect that you confused panels b) and c). If not, this comment might have severe implications. Please verify.*

We appreciate the reviewer's pointing this out. The panel swap mentioned in Fig. 2 was definitely a mistake, which we have now corrected. The related buttressing number calculations, however, were / are correct and unaffected by this. Consequently, the mistake in this figure was isolated and did not / does not propagate to any of the discussion or conclusions in the rest of the paper.

*Inconsistent and biased analysis: I certainly appreciate how carefully you have structured the analysis in this manuscript. You clearly state a correlation between the GLF response and the buttressing index in dynamically relevant areas (cf., Sect. 4.2, Fig.4). Thereafter, you show that buttressing changes in the vicinity of the thinning perturbations exhibit a counterintuitive behaviour which is difficult to interpret. Yet, this difficulty seems to have entirely undermined your confidence in the interpretability of this measure. In the abstract, you even condemn the correlation between the GLF and the local buttressing measures as remaining '[...] elusive from a physical perspective'.*
    *This judgment is evoked throughout your manuscript and I somehow feel that I have to take up the cudgels for this metric. First, you show yourself that there can be good correlation (Figs. 4,5,11b). The more I tried to understand the details of your analysis, I have more and more doubts about its robustness. First doubts arose when I read through Sect. 4.3.1. You start by discussing non-local speed-up in the vicinity of the perturbed area (but excluding the centre). Thereafter, you focus on the local-scale buttressing changes within the perturbed area. This seemed inconsistent and this choice biases and discredits the buttressing change measure.*

As noted above, we do eventually conclude that the correlation between local buttressing number and grounding line flux should not be applied in a 'predictive' sense (i.e., to diagnose the difficulty to calculate GLF sensitivity via the much simpler to calculate (local) buttressing number. Our analysis in section 4.3, which has been significantly revised and updated (including updated analysis and discussion of perturbations and the resulting changes that occur *at* the location of perturbations) is consistent with these conclusions. Throughout our revised section 4.3, we have attempted to clarify and emphasize the fundamental inconsistencies we find between the impacts of (1) local (at the grid cell) perturbations on various physical quantities, including the buttressing number, versus (2) changes in areas neighboring the immediate perturbation, versus (3) domain-integrated changes in buttressing and ice flux at the grounding line, and to more clearly tie the findings from this section of the paper to the broader discussion

and conclusions. In general, we show that, on the ice shelf, changes in buttressing at and immediately neighboring to perturbation locations are generally not consistent with our broader understanding for how buttressing works and also not consistent with the changes observed by the integrated ice shelf / ice sheet system explored here (i.e., local perturbations (reductions) in ice shelf thickness *reduce* overall ice shelf buttressing, which in turn *increases* overall ice flux across the grounding line).

*Initially, I was willing to accept this judgment but then I realised that the same counter-intuitive response is seen in the principal strain-rate components (Fig.7e and f). These also indicate compression within the perturbed area (and slightly beyond). Consequently, you also need to dismiss the usefulness of this measure. This is too much of a stretch for me. I simply think that your analysis should consistently avoid areas close to the perturbations. To substantiate my view, I want to briefly explain the 1st principal buttressing or strain-rate changes in Fig7 e and g. After the perturbation, you clearly get less buttressing and increased extension upstream and downstream (in-flow direction) of the affected area. Sideways, but still along the 1st principal direction, these effects result in increased buttressing and compression (similar to a bottleneck effect). This explanation seems reasonable. I therefore strongly urge you to moderate and adjust your assessment of the buttressing metric, accordingly*

This is a difficult argument to follow because, by nature, the buttressing number calculations are *local* in nature. It's hard to support their use on the basis of physical arguments if one cannot understand and connect local changes in buttressing to the broader changes in buttressing that control overall flux across the grounding line. Nevertheless, we go through a detailed analysis in our revised section 4.3 (and related Fig. 6) where we attempt to connect the local and neighboring impacts of perturbations on the shelf (including their impacts on buttressing number) to the broader changes in buttressing experienced by the entire ice shelf and their impacts on grounding line flux. While we can provide a fairly detailed narrative for *what* happens when a perturbation is applied to the ice shelf, we still lack a convincing physical understanding for *why* it happens. That is, why should the grounding link flux sensitivity -- an integrated quantity -- be correlated or adequately characterised by a locally calculated buttressing number on the ice shelf? We cannot confidently answer that question here, which gives us great hesitation in blindly applying these correlations. Moreover, our findings that many other easily derived physical quantities (some trivial, e.g. ice thickness) also correlate well with grounding line flux suggest that there may be no direct physical connection between these two quantities that would support their broader use (the correlations could be spurious or fortuitous, as discussed in newly added parts of Section 4.3.4 and discussion and Table S1 in the Supplementary Material). Regardless, we clearly show that these same correlations become weak and unconvincing when applied to realistic ice shelf domains. We would be happy for someone else to carry on with trying to further understand and justify the use of local buttressing numbers as part of ongoing work, but that is not the goal of our paper. The reviewer is suggesting that we come up with a better definition for, calculation of, and understanding of a buttressing number that takes non-local factors into account. This is a laudable goal, but again, is well outside the stated aims and scope of this paper.

*A main motivation for why I raise this point is that many ice-flow models are not capable of an adjoint-based evaluation. It would therefore be constructive, if you could give some advice on how to best evaluate the local buttressing metric wrt. the GLF sensitivity. You nicely show that there is a correlation.*

We fully appreciate this and, as stated above, it was a primary motivation when we initially undertook this study. Unfortunately, we cannot advocate further for the application of this method for the reasons argued above.

*Your strategy to introduce a buffer zone around the grounding line is valuable (it is anyhow clear that these regions are important for the GLF sensitivity). From Fig.4, I think that areas with negative buttressing values should also be excluded (gives more flexibility than prescribed masking). So you could give some advise on how this metric can still be useful.*

We have updated and improved the discussion of the necessary "buffer zone" in the revised manuscript. Specifically, we now introduce a more quantitative way of calculating this buffer zone based on the ratio of shear to normal stress (Section 4.1 and Fig. S2). Unfortunately, this does nothing to address the fundamental problems of needing a buffer zone in the first place; 1) this removes many of the most sensitive areas from consideration, and 2) when applied to realistic ice shelves, one is limited to a small area of the ice shelf if strong correlations are of interest.

*Moreover, you should emphasise that if the interest is in the GLF sensitivity, the buttressing metric should be computed in p1/flow direction as against Furst et al. (2016). This comment further implies that you might want to reconsider the structure of the document: I suggest that you start with the GLF sensitivity of Reese et al. (2018). Then you could show that the adjoint-based approach gives equivalent results. Afterwards, you might want to assess the utility of the buttressing metrics (advice, limitations, etc.) to explain the GFL sensitivity*

Indeed, we do clearly argue that buttressing calculated in the p1 and ice flow directions are better for quantifying the GLF sensitivity than the p2 direction, at least for the case where strong correlations are observed. We have not, however, opted to restructure our manuscript as suggested because our current conclusions and recommendations are better supported by the current organization and narrative.

Minimum and maximum speed increase

*It took me a while to get my head around the retrieval of the direction of the minimal and maximal speed increase (L182ff). Although I am very impressed by the distinct peaks in the resultant distribution (Fig.6b), I wonder about its utility in this study. After its presentation, this measure is briefly compared to Gudmundson (2003) and shortly re-raised for the Larsen C*

*setup. It is not discussed nor mentioned in the conclusions. I therefore urge you to re-consider its utility*

This section and the related figures have been removed from the revised version of the paper.

*1. Please reduce the overall amount of footnotes. Sometimes they keep valuable extra information, which should appear in the text.*

We have significantly reduced the number of footnotes by including most of the relevant material in the primary text.

*2. Please introduce a figure of GLF response Nrp and the buttressing values (p1,p2, flow) for Larsen C. It might help you to delineate the area in which the GLF response and buttressing values are correlated.*

This has been added as a third column of panels to a new figure that combines several figures from the original version of the manuscript. This information can now be found in Fig. S6 in the SI.

*L29 The term 'longitudinal stresses' seem to be too narrow here. I would rather speak of 'membrane stresses' following Hindmarsh (2006).*

Thanks for this suggestion. We have updated the manuscript accordingly.

*L42 Delete 'of ice'*

Done.

*L60 Insert comma after parenthesis.*

Corrected.

*L115 This sentence is not true. You do not show the response on the southern/botton part of the MISMIP+ setup.*

What is meant here is that we do analyze the response to perturbations over the entire model domain but we don't conduct perturbation experiments over the entire domain. This is because the response will be symmetric about the centerline. For example, the response of the change in GLF to a perturbation at (x,y)=(480 km, 50 km) will be the same as to a perturbation at (x,y)=(480 km,30 km), just mirrored about the ice stream / shelf centerline.

*L118 As in the original study by Reese et al. (2018), I do not understand the meaning of P. You say it is the local mass change associated with the perturbation. So it should be rather constant*

*(despite element size variations on Larsen C). Units should be m 3 . The GL flux change R is however in units of m 3 /yr. I do not understand how Nrp can then be dimensionless. I think that I misunderstood the meaning of P. Please explain in more detail.*

We have added more information to this section of the paper to clarify the units on both R and P. In Reese et al. (2018), it is implicit that the time period of interest is one year (according to personal communications). Therefore, P should have units of $m^3$, which are the same units as R. We have explicitly stated that the units of R and P are both in $m^3$ (in which case their ratio is non-dimensional).

*L169 You must have noticed the dip in the correlation with the p1-buttressing (Fig. 5a). So the best correlation occurs at ±25∘ . With respect to the flow direction, the optimal correlation is ~100∘ turned (counterclockwise, Fig. 5b). Your statement in this line does note entirely hold.*

We maintain that this statement is correct: if we move the curve in Fig. 5b to the right by around 50 degrees, the point with the best correlation in Fig. 5b moves to 150 deg, similar to the local maximum in Fig. 5a. The point with the second best correlation in Fig. 5b is shifted to ~210 deg, i.e., 30 deg, corresponding to the second local maximum in Fig 5a.

*L173 You envoke the notion of an overall best buttressing metric. I do not think that this exists as such. It will depend on the spatial focus which can be the GL, central areas of the ice shelf or the calving front. Please remove this notion of a best metric.*

The notion of a "best" metric is not ours but comes from the previous work of Fürst et al. (2016). We state this clearly in our paper. We're not supporting its use or definition here. To avoid confusion, we have changed "best" to "good" in this sentence

*L194-L207 Prior to this section, you focus on the speed-up signal 'among neighbouring cells' (L182-L194). In this section, you then discuss buttressing changes within the perturbed areas. This seems inconsistent. From Fig. 7g and h, I think you can extract a meaningful, aggregated index for buttressing changes excluding the perturbation centre. Upstream of the perturbation (in flow or p1 direction), the buttressing decreases with highest decrease close to the perturbed area. This inconsistent treatment therefore seems deliberate and strongly biases your interpretation. This bias leads to harsh judgments of the buttressing metric in the subsequent two sections, which are, in my opinion, note well justified. Please stay more objective. You also show the strain rates fields in the principal direction which also show overall compression within the perturbed zones. You do no discredit the usefulness of these values either.*

We have updated the analysis and discussion in this entire section, including a focus on the impacts of perturbations exactly at the grid cells where perturbations are applied. As noted above, we agree that one can conduct a careful analysis of a single perturbation in order to understand how, overall, that perturbation leads to the broader changes in buttressing that are expressed as changes in GLF. However, we still lack a detailed understanding for how this

cause-and-effect is physically connected to the concept of a locally calculated buttressing number. We also show (in a new section in the SI) that similarly strong correlations exist between GLF and other physical quantities, some of which are unrelated to buttressing or buttressing number. This, and more importantly, the lack of strong correlations when exploring realistic ice shelf domains leads us to abandon further investigation of the utility of this method as a proxy for understanding GLF sensitivity.

*L225-L238 This paragraph judges the results and it is therefore better located in the discussion conclusion. I also sense some redundancy.*

This section (4.3.3), which has been significantly revised, is a necessary summary of our findings from the detailed analysis conducted in the sections immediately above it. Further, it is a necessary transition from discussion of the idealized MISMIP+ test case domain to the more realistic Larsen C domain.

*L273-L289 This paragraph presents methodology so it should appear earlier (not as a sub-section of the Results).*

While this change would make our paper more closely follow the strict formatting of a standard research paper (e.g., introduction, methods, results, conclusions) we think that the overall readability would suffer as a result. Further, a number of other reviewer comments indicate that the paper and interpretation would be easier to follow if this strict partitioning is avoided. Therefore, we opt to keep the formatting of these sections as they currently are.

*Fig.1 Poor figure quality. Missing overview figure for localisation of Larsen C. What did you do about Bawden Ice Rise? Could you also show the observed velocity magnitude on Larsen C. Please indicate in the figure that the velocities you show, present the state after the relaxation (you only mention this in the text L105).*

The location of Larsen C has been added to the figure. We have also added Figure S1 (to the Supplementary Material) showing the comparison between modeled and observed ice surface speeds on Larsen C. With respect to Bawden Ice Rise, we have looked into this in some detail and it appears that it is a small feature that does not show up in our domain due to our initial data interpolation onto a mesh [with a minimum resolution of approximately the same size as this feature](). We thank the reviewer for pointing this out, as it is something we will pay closer attention to including in future meshes.

*Fig.2 In the caption you speak about 'perturbation points'. The perturbation does not affect a single point but an entire patch.*

We now refer to these as "perturbed grid cells" instead of "perturbation points".

*I would use different colours for the response number and buttressing metrics.*

While we tried out multiple colorbars for the different panels in Fig. 2 (panel a vs. panels b, c, and d), we ultimately decided to keep them the same. This is primarily because it is then much easier to compare the spatial pattern of the GLF response number with the spatial patterns of the buttressing numbers in different directions (i.e., making a qualitative comparison by "eyeball").

*Why do you get negative response numbers for perturbations next to the grounding line?*

We observe negative adjoint sensitivities only for cells intersecting the grounding line. For those cells, changes in thickness directly affect the grounding line position/length and the thickness over the GL, which could lead to negative responses. We have added a note to the Fig. 2 caption on the topic of negative response number. This topic is also discussed in the 4th paragraph of Section 4.5 (starting around line 331).

*Figs.11&12 I would try to merge these figures. Panels (a) can be placed as an inset into panels (b).*

As suggested, we have merged Figs. 11 and 12. They have also been moved into the SM (currently as Fig. S6).

---

## Referee Report (RR1)

Summary and comments on the revised manuscript entitled

**Diagnosing the sensitivity of grounding line flux to changes in sub-ice shelf melting**

initially presented on 11.02.2020

by

T. Zhang et al.

**General**

I want to congratulate the authors to this very well written and well structured manuscript, which is rich in content and of high scientific quality. The conclusions drawn in this article will rectify the view of glaciologists on locally defined measures of the ice-dynamic state of ice-shelves with regard to grounding line flux (GLF) response. Their recommendations in terms of assessing the GLF sensitivity are very useful for improving future assessments. The authors succeeded in resolving my main initial concerns and they moderated their assessment on the local measures. As it stands, I recommend this manuscript for publication in *The Cryosphere* after some few technical corrections have been addressed.

**Techincal comments**

**L128** By stating that $R$ is quantified as the 'change in the GLF over a year due to a perturbation in the thickness', I was initially confused and thought that you conduct transient simulations for a year. Yet you clearly state above that you quantify the instant response. Anyhow, I would reformulate this part to avoid confusion. You only need to mention this time period to obtain a non-dimensional number.

**L281** You missed to specify the components which enter the correlation mentioned here.

**L368** Doubling of 'only'.

**Fig. 3** I do not see the necessity to invoke the linear regression analysis in the caption here. Initially it confused my interpretation of the figure.

**Fig. 3** In the caption, you distinguish between 'modeled' and 'predicted' values for $N_{rp}$ but I am not sure how you distinguish them in the panels. I suspect the two shades of blue indicate these two categories. Please amend.

**Fig. 4** Same comments as for Fig. 3.

**Fig. S2** $n_{rp} \longrightarrow N_{rp}$

---

## Author Response (AR2)

**Response to review 1**

*The manuscript has been improved greatly and is not in a much better shape.*

*Personally, I still find that the search for a correlation between locally derived buttressing numbers within an ice shelf and integrated grounding-line flux is a bit beside the point. There will be some correlation because for confined ice-shelves local changes in ice thickness impact both grounding-line flux and the local measure of buttressing. But maybe this is just a question of how to think about these processes. I like to think about the thickness affecting the stresses. The fact that we then many have a correlation between stresses at two different locations, or between stresses where the thickness perturbation was applied and GL flux, is somehow less fundamental to me. But everyone should be entitled to have their own favourite way of thinking about these things. The bottom line is that using locally defined buttressing numbers within an ice shelf as a diagnostic for impacts on GLF is not a particularly sound idea and, as the authors find, any such relationship is tenuous at the best.*

We thank the reviewer for his highly constructive comments. To make our points more clear, we change the sentence "This and the fact the correlation is generally much poorer when applied to realistic ice shelf domains motivates us to seek an alternative approach " in the abstract to "This and the fact the correlation is generally much poorer when applied to realistic ice shelf domains motivates us to seek an alternative approach for predicting changes in grounding line flux ".

*The way the paper is written at the moment I feel there is a potential for some confusion. As far as I can see there are two main points in the paper. One relates to the usefulness of local buttressing numbers within an ice shelf as a diagnostic for impact on GL flux (bad idea), the other one is a purely methodological point: i.e. how to compute the impact of thickness perturbations on GL flux in a computationally efficient way. This second main point of the paper (use the adjoint method to speed up things) has nothing to do with the first point. But as the paper is written at the moment, it might appear to a reader that is not too careful, that the adjoint-based sensitivity calculation somehow links to the first point, or that it even offers a solution to the (lack) of correlation between locally calculated buttressing and GL flux.*

To make this point more clear, we now change the following sentence in Section 4.5, "This motivates our investigation of a wholly different approach, which provides a GLF sensitivity map analogous to that from Reese et al. (2018).", to "This motivates our investigation of a wholly different approach, which provides a GLF sensitivity map analogous to that from Reese et al. (2018), instead of seeking for a simple buttressing number indicator to predict the GLF sensitivity. "

*I don't see how the Rees approach can be considered 'ad hoc'. It's just computationally inefficient. In fact, one could argue that the Rees approach as the advantage that it does not have the inbuilt linearization of the adjoint method. The adjoint approach is much better at estimating the linear response than the finite-differences methods. It is both more accurate, and faster. But the (first-order) adjoin approach use by the authors cannot be used to estimate the range (amplitude) of the linear response as done by Rees. Also, in this particular example the adjoint method will not really make that much of a difference unless one only want to know the integrated impact on GL flux. As soon as one wants to calculate the perturbation at each and every point along the grounding line (as done by Rees) I suspect the computational efforts using the finite-differences and the adjoint methods will become similar.*

We now remove "ad hoc" in the sentence "Thus, despite the added complexity in its computation, the adjoint-based method provides significant advantages over the simpler but more ad hoc(i.e., perturbation-based) analysis methods

discussed above." The new sentence is "Thus, despite the added complexity in its computation, the adjoint-based method provides significant advantages over the simpler perturbation-based analysis methods discussed above."

*Not sure about this as I'm not that familiar with the notion, but is (6) not generally true whenever atm. pressure is neglected?*

Equation (6) is true when we neglect atmosphere pressure (stress-free condition) on the surface. See Equation (14) in Perego et al. (2012).

*I feel that the discussion about ice-shelf buttressing should reference the papers by:*
*Pegler, S. S.: Suppression of marine ice sheet instability, J. Fluid Mech., 857, 648–680, doi:10.1017/jfm.2018.742, 2018.*
*Pegler, S. S.: Marine ice sheet dynamics: the impacts of ice-shelf buttressing, J. Fluid Mech., 857, 605–647, doi:10.1017/jfm.2018.741, 2018.*
*Haseloff, M. and Sergienko, O. V.: The effect of buttressing on grounding line dynamics, J. Glaciol., 64(245), 417–431, doi:10.1017/jog.2018.30, 2018.*
*This could, for example, be done early in the paper (for example at the end of line 25) but there are number of other places were this could be added. These three papers are quite important addition to the literature.*

These three new references are now added.

*Citation to Cornford 2020 should be undated to refer to the final published version.*

Updated.

References:
Perego, M., Gunzburger, M., & Burkardt, J. (2012). Parallel finite-element implementation for higher-order ice-sheet models. Journal of Glaciology, 58(207), 76-88. doi:10.3189/2012JoG11J063

**Response to review 2**

*General*

*I want to congratulate the authors to this very well written and well structured manuscript, which is rich in content and of high scientific quality. The conclusions drawn in this article will rectify the view of glaciologists on locally defined measures of the ice-dynamic state of ice-shelves with regard to grounding line flux (GLF) response. Their recommendations in terms of assessing the GLF sensitivity are very useful for improving future assessments. The authors succeeded in resolving my main initial concerns and they moderated their assessment on the local measures. As it stands, I recommend this manuscript for publication in The Cryosphere after some few technical corrections have been addressed.*

We thank the reviewer for the support of publishing this manuscript on TC.

*Techincal comments*

*L128 By stating that R is quantified as the 'change in the GLF over a year due to a perturbation in the thickness', I was initially confused and thought that you conduct transient simulations for a year. Yet you clearly state above that you quantify the instant response. Anyhow, I would reformulate this part to avoid confusion. You only need to mention this time period to obtain a non-dimensional number.*

We change "where R is the change in the GLF over a year due to a perturbation in the thickness at a single grid cell" to "where R is the volume change by GLF over a year due to a perturbation in the thickness at a single grid cell".

*L281 You missed to specify the components which enter the correlation mentioned here.*

We now change this sentence to "Further, we show in the Supplementary Material and Table S01 that the correlation between Nb and Nrp may be spurious"

*L368 Doubling of 'only'.*

The first "only" is removed.

*Fig. 3 I do not see the necessity to invoke the linear regression analysis in the caption here. Initially it confused my interpretation of the figure.*

We change "linear regression" to "relation".

*Fig. 3 In the caption, you distinguish between 'modeled' and 'predicted' values for Nrp but I am not sure how you distinguish them in the panels. I suspect the two shades of blue indicate these two categories. Please amend.*

We rephrase it as "Modeled Nrp versus buttressing number Nb calculated along..."

*Fig. 4 Same comments as for Fig. 3.*

Corrected.

*Fig. S2 nrp -→ Nrp*

Corrected.